



# The novel GOME-type Ozone Profile Essential Climate Variable (GOP-ECV) data record covering the past 26 years

Melanie Coldewey-Egbers[1], Diego G. Loyola R.[1], Barry Latter[2,3], Richard Siddans[2,3], Brian Kerridge[2,3], Daan Hubert[4], Michel van Roozendael[4], and Michael Eisinger[5]

[1]Remote Sensing Technology Institute, German Aerospace Centre (DLR), Wessling, Germany
[2]Remote Sensing Group, STFC Rutherford Appleton Laboratory, Chilton, United Kingdom
[3]National Centre for Earth Observation, STFC Rutherford Appleton Laboratory, Chilton, United Kingdom
[4]Royal Belgian Institute for Space Aeronomy (BIRA-IASB), Brussels, Belgium
[5]European Space Agency - European Centre for Space Applications and Telecommunications (ESA-ECSAT), Oxford, United Kingdom

**Correspondence:** Melanie Coldewey-Egbers (Melanie.Coldewey-Egbers@dlr.de)

**Abstract.** We present the GOME-type Ozone Profile Essential Climate Variable (GOP-ECV) data record covering the 26-year period from July 1995 until October 2021. It is derived from a series of five nadir-viewing ultraviolet-visible(-near-infrared) satellite instruments of the GOME-type, including GOME/ERS-2, SCIAMACHY/ENVISAT, OMI/Aura, GOME-2/MetOp-A, and GOME-2/MetOp-B, which are merged into a single coherent long-term time series. It provides monthly

mean ozone profiles at a spatial resolution of $5° \times 5°$ latitude by longitude. The profiles are given as partial columns for 19 atmospheric layers ranging from the surface up to 80 km. The underlying profile retrieval algorithm is the Rutherford Appleton Laboratory scheme, which has sensitivity to both tropospheric and stratospheric amounts of ozone. The merged profile record has been developed by the German Aerospace Center (DLR) in the framework of the European Space Agency's Climate Change Initiative+ (ESA-CCI+) ozone project (Ozone_CCI+). Profiles from the individual instruments are first harmonized

through careful inspection and elimination of inter-sensor deviations and drifts and then merged into a combined record. In a further step, the merged time series is harmonized with the GOME-type Total Ozone Essential Climate Variable (GTO-ECV) data record, which is based on nearly the same satellite sensors. GTO-ECV possesses an excellent long-term stability and with the homogenization an improvement of the robustness and stability of the merged profiles can be achieved. For this purpose, an altitude-dependent scaling is applied that utilizes ozone profile Jacobians obtained from a Machine Learning approach. We

found that climatological ozone distributions derived from the final GOP-ECV data record agree with spatial and temporal patterns obtained from other long-term data records.

## 1   Introduction

Ozone as a key constituent of the atmosphere absorbs most of the damaging incoming solar radiation and thereby protects and benefits life on Earth. Severe thinning of the ozone layer, most pronounced over Antarctica and caused by catalytic chemistry

involving chlorofluorocarbons (CFCs), was observed since the 1980s. Nowadays, evaluating the success and the impact of the Montreal Protocol and its subsequent amendments (United Nations Environment Programme, 2020), which have been



established in order to regulate and to phase out the production of detrimental ozone-depleting substances (ODSs), on the long-term evolution of the ozone layer is important and of major concern (Braesicke et al., 2018; SPARC/IO3C/GAW, 2019; Hassler et al., 2022). Recent studies show that the substantial decline in ozone amounts could be banned (Hassler et al., 2022), but significant increases are still restricted to just a few regions on the globe, e.g., for total ozone columns in the Southern Hemisphere (Coldewey-Egbers et al., 2022; Hassler et al., 2022; Weber et al., 2022). For vertically resolved ozone the increase is limited to certain altitude levels, i.e. the upper stratosphere outside of the polar regions (Braesicke et al., 2018; SPARC/IO3C/GAW, 2019; Sofieva et al., 2021; Godin-Beekmann et al., 2022; Hassler et al., 2022). On the other hand, total ozone trends are close to zero or not significant in the Northern Hemisphere and the tropical region, respectively. For the lower stratosphere, there is also some indication of a small though uncertain decrease in ozone amounts in the tropics and the middle latitudes of the Northern Hemisphere (Hassler et al., 2022).

Any assessment of long-term ozone trends and their associated uncertainties as well as investigations of inter-annual variability in ozone and a possible impact of climate change require reliable underlying data sets. Preferably they span several decades and provide (near-)global coverage. Since the 1970s satellite observations possess this potential, although single missions are generally only operating for a relatively short period (about 5–15 years). Thus, merging measurements from multiple sensors into an homogeneous long-term composite is necessary. Over the past one and a half decades several methodologies addressing this task have been developed (Tummon et al., 2015; SPARC/IO3C/GAW, 2019). However, combining two or more different records requires comprehensive consideration of various aspects such as different vertical and horizontal resolutions, measurements that are made at different times of the day, different vertical coordinate systems, or diverse calibration and retrieval algorithms. In some cases spatial and/or temporal sampling patterns for individual sensors can change over their lifetimes (McPeters et al., 2013). Moreover space-based instruments commonly suffer from optical throughput degradation that can affect the long-term stability (Miles et al., 2015). All these factors need to be taken into account before merging, since they may induce discontinuities, artificial drifts, or sudden jumps, which can hence distort the estimated decadal trends (Tummon et al., 2015; Hubert et al., 2016).

A number of merged data records of vertically resolved ozone amounts has been created in the past years. The ones with high vertical resolution are based on limb and occultation instruments, e.g., the Global Ozone Chemistry and Related Trace Gas Data Records for the Stratosphere (GOZCARDS; Froidevaux et al., 2015), the Stratospheric Water and Ozone Satellite Homogenized dataset (SWOOSH; Davis et al., 2016), the Stratospheric Aerosol and Gas Experiment–Optical Spectrograph and Infrared Imaging System–Ozone Mapping and Profiler Suite record (SAGE-OSIRIS-OMPS; Bourassa et al., 2014), the Stratospheric Aerosol and Gas Experiment–Scanning Imaging Spectrometer for Atmospheric Chartography–Ozone Mapping and Profiler Suite record (SAGE–SCIAMACHY–OMPS; Arosio et al., 2019), or the Merged Gridded Dataset of Ozone Profiles (MEGRIDOP; Sofieva et al., 2021). Typically, the merging procedures rely on selecting one sensor as a reference. The others are then adjusted with respect to the reference based on comparisons during overlap periods. Merging is either based on absolute values (Froidevaux et al., 2015; Davis et al., 2016) or on deseasonalized anomalies (Bourassa et al., 2014; Sofieva et al., 2021). On top of that, Ball et al. (2017) have developed the Bayesian Integrated and Consolidated (BASIC) composite, which itself is composed of existing merged ozone profile records (amongst others GOZCARDS and SWOOSH). Their novel





approach uses Bayesian methods in order to identify and to finally remove spurious features such as jumps or artificial drifts in the underlying data sets.

In addition to limb- and occultation-based data sets two merged ozone profile records constructed from nadir-viewing sensors
are available, namely the SBUV-MOD (Solar Backscatter UltraViolet Merged Ozone Data Set; Frith et al., 2014) generated by the National Aeronautics and Space Administration (NASA) and the SBUV-COH (Wild and Long, 2023) produced by the National Oceanic and Atmospheric Administration (NOAA). They consist of a series of BUV, SBUV, and SBUV-2 instruments as well as the OMPS sensor onboard the Suomi National Polar-orbiting Partnership (S-NPP) platform. Ozone profiles are retrieved using a common algorithm (Version 8.6; McPeters et al., 2013), but different calibration techniques are applied in
order to harmonize the measurements.

In this paper we present the novel GOME-type Ozone Profile Essential Climate Variable (GOP-ECV) data record that is based on observations from a series of nadir-viewing ultraviolet-visible(-near-infrared) (UVN) satellite sensors of the GOME-type (Burrows et al., 1999). It is the first European merged profile data record of this kind and has been developed in the framework of Phase-2 of the European Space Agency Climate Change Initiative (ESA-CCI+) ozone project (Ozone_CCI+).
The main objective of this work was not only to merge the measurements from individual nadir sensors into an homogeneous data record covering two and a half decades (1995–2021), but also to achieve internal consistency (in terms of the total ozone amount) with the well-established GOME-type Total Ozone Essential Climate Variable data record (GTO-ECV; Coldewey-Egbers et al., 2022, and references therein), which is generated by DLR. From the alignment w.r.t. GTO-ECV we expect a positive impact on the long-term stability of GOP-ECV compared to a merely combined nadir profile record. Both GOP-ECV
and GTO-ECV are based on nearly the same satellite sensors (see Sec. 2). Moreover, we can take advantage of (1) the same ozone profile retrieval algorithm that is applied to all sensors (Miles et al., 2015, and see also Sec. 2.2), (2) a common coordinate system and the same units for all datasets to be merged, (3) sufficiently long overlap periods with the chosen reference OMI (Ozone Monitoring Instrument; Levelt et al., 2018), that enable a robust analysis of inter-sensor biases and drifts, and (4) good spatial and temporal coverage and good horizontal and temporal resolution. The latter allows us to generate a latitudinally and
longitudinally resolved data record, which is in particular important for the evaluation of regional height-resolved ozone trends (Arosio et al., 2019; Sofieva et al., 2021; Coldewey-Egbers et al., 2022).

In addition to investigations of changes in the profile, of particular interest is the analysis of the long-term evolution of ozone amounts in the troposphere. Tropospheric ozone acts as an effective greenhouse gas and is a severe air pollutant. Its distribution is highly variable in space and time. Since information about the ozone content in the lowermost atmospheric layer
can be retrieved from UVN sensors (Miles et al., 2015), GOP-ECV enables us to assess regional and global long-term changes, which will contribute to a better understanding of trends in total ozone columns (Hassler et al., 2022).

The present paper focuses solely on the description of the approach that has been developed to generate the GOP-ECV data record. In a subsequent study, results of the geophysical validation using ground-based data, e.g., ozone sondes, and of comparisons with other satellite-based records as well as first climate applications will be presented. The outline of this paper
is as follows. Section 2 contains an overview of the satellite sensors and the ozone profile data sets, which are included in GOP-ECV. Moreover, we briefly describe the total ozone data record GTO-ECV. In Sec. 3 we introduce the harmonization and





merging approach applied to the individual profiles followed by a detailed description of the homogenization procedure w.r.t. the GTO-ECV data record (Sec. 4). In Sec. 5, we discuss the climatological ozone distribution at selected atmospheric layers and in Sec. 6, we present the summary and outlook.

## 2 Data

### 2.1 Satellite sensors

For the generation of the GOP-ECV climate data record of ozone profiles measurements from five nadir-viewing ultraviolet-visible-near-infrared (UVN) satellite sensors are combined, viz. GOME (Global Ozone Monitoring Experiment) onboard ERS-2 (second European Remote Sensing satellite), SCIAMACHY (Scanning Imaging Spectrometer for Atmospheric Chartography) on ENIVISAT (Environmental Satellite), two GOME-2 sensors on MetOp-A and MetOp-B (Meteorological Operational satellites A and B), respectively, and OMI (Ozone Monitoring Instrument) onboard Aura. Table 1 provides an overview of the instruments, their lifetimes and equator crossing local times and indicates the periods for which data is included in the merged record.

All platforms fly in sun-synchronous, near-polar low earth orbits. ERS-2, ENVISAT, and MetOp cross the equator in the mid-morning between 09:30 LT and 10:30 LT (descending node, see also Table 1), whereas the Aura satellite is in an early afternoon orbit (13:30 LT, ascending node). The various local times of the day at which the ozone profile measurements are conducted can lead to systematic biases between the individual instruments caused by the diurnal cycle in both the troposphere and the stratosphere. The diurnal cycle is well pronounced and has large variations in the upper stratosphere and the mesosphere (Prather, 1981; Pallister and Tuck, 1983). Peak-to-peak amplitude is about 5–15% depending on the latitude, altitude, and season. Moreover, the upper stratosphere and the mesosphere exhibit different daily patterns, i.e. maxima and minima occur at different daytimes (Parrish et al., 2014). Notable variation was also found in the middle stratosphere (Sakazaki et al., 2013). Thus, when assembling multiple ozone profile data records these diurnal variations need to be considered since they have the potential to introduce significant biases. For this reason, the merging approach we apply (see Sec. 3) is based on de-seasonalized anomalies in order to remove those possible biases induced by different overpass times.

Due to the various swath widths (see Table 1) global coverage is achieved after three days for GOME, after six days for SCIAMACHY (owing to the alternating limb and nadir measurements), almost daily for GOME-2 (with the exception of only the tropics), and daily for OMI. The polar regions are characterized by multiple views per day. Additionally, the sizes of the ground-pixels are considerably different for the individual instruments. They vary from $320 \times 40\,\mathrm{km^2}$ for GOME to $13 \times 24\,\mathrm{km^2}$ for OMI. More details on the sensor characteristics can be found in Burrows et al. (1999), Bovensmann et al. (1999), Levelt et al. (2018), and Munro et al. (2016).





**Table 1.** Overview of individual nadir-viewing satellite sensors included in GOP-ECV

| Instrument/Platform | Period of operation | Period in GOP-ECV | Overpass | Swath width | Reference |
| --- | --- | --- | --- | --- | --- |
| GOME/ERS-2 | 06/1995 – 07/2011 | 07/1995 – 12/2002 | 10:30 LT[a] | 960 km | Burrows et al. (1999) |
| SCIAMACHY/ENVISAT | 08/2002 – 04/2012 | 09/2002 – 12/2004 | 10:00 LT | 960 km | Bovensmann et al. (1999) |
| OMI/Aura | 10/2004 – today | 10/2004 – 10/2021 | 13:30 LT | 2600 km | Levelt et al. (2018) |
| GOME-2/MetOp-A[b] | 01/2007 – 11/2021 | 01/2007 – 12/2016 | 09:30 LT | 1920 km[c] | Munro et al. (2016) |
| GOME-2/MetOp-B[d] | 01/2013 – today | 01/2015 – 10/2021 | 09:30 LT | 1920 km | Munro et al. (2016) |

[a]LT = Local time at the equator; [b]in the following we refer to GOME-2 onboard MetOp-A as GOME-2A; [c]reduced to 960 km from 07/2013 onward; [d]in the following we refer to GOME-2 onboard MetOp-B as GOME-2B.

## 2.2 Input ozone profile products

Ozone profiles from the nadir satellite sensors described in the previous section are retrieved using the Rutherford Appleton Laboratory (RAL) scheme, that is a sequential three-step process and described in detail in Miles et al. (2015). Retrieved profiles are provided on a fixed vertical pressure grid with 20 levels from the surface up to 80 km. In step one, the ozone profile is derived from Sun-normalized radiances in selected wavelength intervals of the Hartley band in the spectral region 265–307 nm. This range mainly contains information on ozone in the stratosphere. Next, the surface albedo for each satellite ground-pixel is retrieved from Sun-normalized radiances in the 1 nm-wide band 335-336 nm in step two of the procedure. Finally, in step three, information on ozone content in the troposphere and the lower stratosphere can be retrieved from the spectral structures in the ozone Huggins band through exploitation of their dependence on temperature. For OMI version 2 of the retrieval scheme is used, whereas version 3 is used for the other four sensors.

Monthly averaged Level-3 profile products generated from the pixel-based Level-2 products described above, were taken from the Copernicus Climate Data Store (CDS) archive. Depending on the sensor and/or the time period, versions 0006, 0007, or 0008 are utilized. For the construction of the Level-3 products, the Level-2 data were filtered, e.g. with respect to cloud fraction or solar zenith angle, according to the criteria given in Keppens et al. (2018, their Table 3). Monthly mean profile information is provided on a $1° \times 1°$ latitude-longitude grid and a description of the gridding algorithm as well as corresponding geophysical validation results can be found in Keppens et al. (2018). For the generation of the GOP-ECV data record, we decided to produce $5° \times 5°$ latitude-longitude averages, which then serve as input to the harmonization and merging approach. Moreover, for our approach we use the partial ozone column amounts provided for the 19 layers between the retrieval pressure levels.

Recently, Pope et al. (2023) have generated a merged data record of lower tropospheric column ozone (surface–450 hPa) inferred from RAL Level-2 ozone profile products described above from GOME, SCIAMACHY, and OMI between 1996 and 2017. They investigated the long-term spatiotemporal variability and evolution and found significant increases in the tropical and sub-tropical region.





### 2.3 GTO-ECV total ozone data record

The GTO-ECV data record of total ozone columns (TOCs) is a merged climate data product combining measurements of a series of seven nadir-viewing satellite sensors. In addition to the five sensors introduced in the previous section (GOME, SCIA-MACHY, OMI, GOME-2A, and GOME-2B), which will constitute the new merged profile record GOP-ECV, measurements performed with TROPOMI (Tropospheric Monitoring Instrument) onboard the Sentinel-5 Precursor platform (from May 2018 onward) and with GOME-2 onboard MetOp-C (from July 2019 onward) are incorporated in GTO-ECV as well. For a detailed

description of this total ozone data record and the corresponding merging approach developed for the total columns we refer to Loyola et al. (2009), Loyola and Coldewey-Egbers (2012), Coldewey-Egbers et al. (2015, 2020, 2022), and Garane et al. (2018). In the following we provide a brief overview of the main characteristics of GTO-ECV.

GTO-ECV covers the period from July 1995 through December 2023. As part of the EU Copernicus Climate Change Service (C3S) ozone project it is extended on a quasi-operational basis and it is freely available from the Copernicus Climate Data Store

(CDS; https://cds.climate.copernicus.eu, last access: 28 July 2022). Monthly mean total ozone columns on a spatial grid of $1° \times 1°$ and corresponding error estimates are provided.

The total ozone columns, which are included in GTO-ECV, are retrieved from the seven nadir satellite sensors using the GOME Direct Fitting version 4 (GODFIT_V4) algorithm (Lerot et al., 2014; Garane et al., 2018). Using the same algorithm for all sensors leads to a basically high inter-sensor consistency of the retrieved total ozone columns, which is generally within

1% for latitudes between 50°N and 50°S (Garane et al., 2018). From the individual level-2 products corresponding daily and monthly level-3 products are calculated, which form the basis of the merged product. In order to harmonize the single products, the OMI data serve as a reference product and the other sensors are adjusted to this reference based on comparisons during overlap periods. Finally the individual data sets are merged into a single product.

Garane et al. (2018) presents the outcome of the geophysical validation of GTO-ECV against the ground-based reference,

which comprises Dobson, Brewer and zenith-sky instruments. A very good overall agreement with 0.5% to 1.5% peak-to-peak amplitude and a negligible long-term drift well below 1% per decade were found. Several studies related to the evaluation of the long-term evolution of total ozone including decadal trends or interannual variability broadly demonstrated the usefulness of GTO-ECV (Coldewey-Egbers et al., 2014; Chipperfield et al., 2018; Weber et al., 2018; Eleftheratos et al., 2019; Coldewey-Egbers et al., 2020, 2022; Weber et al., 2022). Moreover, the results of the validation corroborate the potential merit of GTO-

ECV for improving the consistency and long-term coherence of the merged ozone profiles.

### 3 Merging approach for ozone profiles

The method that is used for generating the merged data record of ozone profiles is very similar to that proposed in Sofieva et al. (2017, 2021), who created various merged ozone profile data sets from limb sensors also in the framework of the Ozone_CCI+ project. As input we use the Level-3 profiles as described in Sec. 2.2. For each of the five satellite sensors we compute absolute



deseasonalized anomalies $\delta_n$ as:

$$\delta_n(i,j,k,t) = \rho_n(i,j,k,t) - \rho_{m,n}(i,j,k), \tag{1}$$

where $\rho_n(i,j,k,t)$ is the monthly mean value for sensor $n$, latitude bin $i$, longitude bin $j$, altitude layer $k$, and month $t$. $\rho_{m,n}(i,j,k)$ is the corresponding climatological mean for sensor $n$ and for month $m$ for each calendar month from January to December. The uncertainties of the deseasonalized anomalies are computed similar to those estimated for the MEGRIDOP

(Sofieva et al., 2021). Their approximation takes into account the uncertainties of the monthly mean gridded ozone profiles $\sigma_n(i,j,k,t)$ and the uncertainties of the corresponding seasonal cycles $\sigma_{m,n}(i,j,k)$ . The uncertainties of the monthly mean gridded profiles $\sigma_n$ are estimated by the standard errors of the means, which are defined as the standard deviations of the monthly samples divided by the square roots of the sample sizes. The uncertainties of the seasonal cycles $\sigma_{m,n}$ are computed following

$$\sigma_{m,n}(i,j,k) = \frac{1}{N_m^2} \sum_{l=1}^{N_m} \sigma_n^2(i,j,k,t_l). \tag{2}$$

$N_m$ is the number of monthly mean values during the selected reference periods (see details given in the next paragraph) for a given month $m$ (January,..., December). The uncertainties of the absolute deseasonalized anomalies are finally estimated as

$$\sigma_{\delta_n}(i,j,k,t) = \sqrt{\sigma_n^2(i,j,k,t) + \sigma_{m,n}^2(i,j,k)}. \tag{3}$$

Since the five instruments do not have one common overlap time (see also Table 1), the seasonal cycles are calculated based

on different reference periods, which cover at least six years and which were set to 1996–2002 for GOME, 2005–2010 for SCIAMACHY, 2005–2020 for OMI, 2007–2016 for GOME-2A, and 2015–2020 for GOME-2B, respectively. Although no common overlap period for all instruments exists, OMI itself has overlap with all the other sensors during different time spans (see also Sec. 2.3). Those overlap periods will be used later on in order to align the individual anomalies.

Figure 1 shows examples of the climatological means, i.e. the seasonal cycles $\rho_{m,n}$ for three spatial bins in the tropics (0–

5° N, (c) and (d)), and the middles latitudes of the Northern and Southern Hemispheres (45–50° N/S, (a), (b), (e), and (f), respectively). The selected longitudinal bin is 0–5° E in all cases, and we show the seasonal cycle for the two layers 27–32 km ((a), (c), and (e)) and 40–43 km ((b), (d), and (f)). In general the seasonal cycles are very similar for all instruments with respect to the location of the maxima and minima and the amplitudes. Largest deviations are observed for the SH middle latitudes ((e) and (f)). Differences can be due to instrumental biases, differences in spatial and temporal sampling, but (as a consequence

of the different periods used) also due to long-term changes in the ozone layer and thus changes in the climatological mean. Usually SCIAMACHY (orange curves) exhibits the largest bias with respect to the other sensors. Except for the lower layer (27–32 km) in the SH middle latitudes (Fig. 1(e)) from April to August, the bias of SCIAMACHY is negative. In the aforementioned spatial bin (Fig. 1(e)), the seasonal cycle retrieved from SCIAMACHY has a much smaller amplitude than the other instruments. Additionally, GOME-2A shows large positive deviations with respect to the other sensors for the layer 40–43 km

in the SH middle latitudes (red curve in Fig. 1(f)). In general, the seasonal cycle strongly depends on the latitude and altitude. In the tropics, a semiannual variation is observed. Going from the lower (27–32 km) to the upper (40–43 km) layer, the cycles reverse concerning the location of the maxima and minima.







**Figure 1.** Seasonal cycles of ozone partial columns for two layers: 27–32 km and 40–43 km in three latitude bands: 45–50° N ((a) and (b)), 0–5° N ((c) and (d)), and 45–50° S ((e) and (f)),respectively, and for five satellite sensors: GOME (blue), SCIAMACHY (orange), OMI (green), GOME-2A (red), and GOME-2B (violet).The errorbars denote the $2\sigma$ standard deviations. For OMI these are additionally highlighted by the green shaded area. Note that the seasonal cycles are computed from different time periods (see text and panel (c). The longitude bin is always 0–5° E.





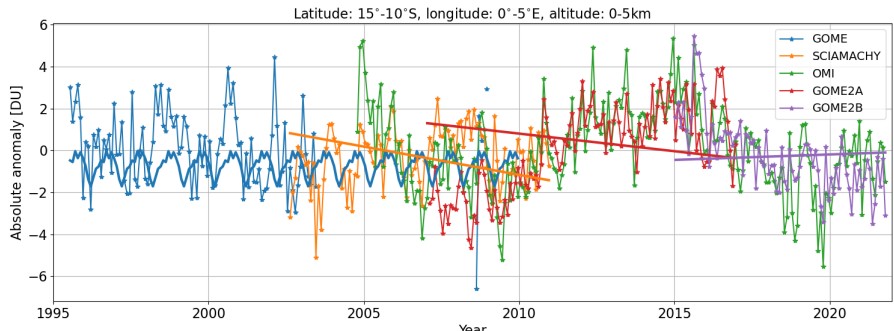

**Figure 2.** Absolute anomalies for GOME (blue), SCIAMACHY (orange), OMI (green), GOME-2A (red), and GOME-2B (purple) as a function of time fom 1995 to 2021 for the 10–15° S, 0–5° E latitude-longitude bin and the lowermost altitude layer (0–6 km). The straight lines for SCIAMACHY, GOME-2A, and GOME-2B (orange, red, and purple) denote the results of the polynomial fits of the offsets between the respective sensor and OMI anomalies obtained from overlap periods (see text). The thick blue curve denotes the corresponding adjustment derived for GOME.

The next step is to align and harmonize the absolute anomalies as computed using Eq. 1 from the different sensors. Deviations between anomalies might be caused by the different reference periods used for calculating the seasonal cycle. On top of that,
these deviations can possibly change over time due to a relative drift between the sensors. Similar to the generation of GTO-ECV (see Sec. 2.3) we use OMI as a reference sensor for the alignment. OMI has sufficiently long overlap periods of at least five years with all other instruments. For the purpose of harmonization, we consider the periods 2005–2009 for GOME, 2005–2010 for SCIAMACHY, 2007–2016 for GOME2-A, and 2015–2020 for GOME-2B, respectively. Offsets w.r.t. the OMI anomalies are computed for each latitude-longitude-altitude-month bin, for which data for the respective sensor and OMI is available. To
the offsets derived for SCIAMACHY, GOME-2A, and GOME-2B, a first order polynomial fit as a function of time is applied to each spatial bin in order to get an estimate of a possible bias and long-term drift drift.

Figure 2 exemplarily shows absolute anomalies for all sensors as a function of time for the 10–15° S, 0–5° E latitude-longitude bin and the lowermost altitude layer. In addition, the straight lines for SCIAMACHY, GOME-2A, and GOME-2B (orange, red, and purple lines, respectively) denote the results of the polynomial fits of the offsets between the respective sensor
and OMI anomalies obtained from overlap periods (see above). For SCIAMACHY (orange) and GOME-2A (red), a clear negative drift was found, whereas for GOME-2B, the drift is slightly positive. In general, this behavior varies with latitude, longitude, and altitude. For the alignment w.r.t. OMI, we then apply the results from the fit as a correction to the anomalies of the particular sensor. The thick blue curve denotes the adjustment derived for GOME, and its estimation is described in the next paragraph.
For GOME it is not possible to calculate offsets during the overlap 2005–2009 for each spatial bin, since GOME lost its global coverage in June 2003 due to a permanent technical failure of the on-board tape recorder. Therefore, we can compute offsets during that period for each layer only for those spatial bins for which measurements are available. From these deviations





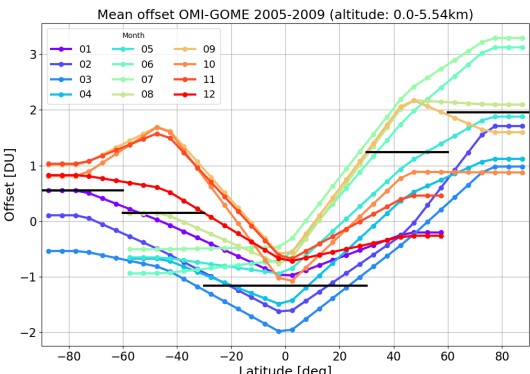

**Figure 3.** Final additive correction values for GOME as a function of latitude and for each calendar month for the lowermost altitude layer (0–6 km). The black horizontal lines denote the annual average correction for each broadband latitude belt: 90–60° S, 60–30° S, 30° S–30° N, 30–60° N, and 60–90° N.

per individual month, at first, we calculate averages for each calendar month from January to December for each available spatial bin, and then we average these monthly ("climatological") means over five broadband latitude bands: 90–60° S, 60–30° S, 30° S–30° N, 30–60° N, and 60–90° N, respectively. In the Northern Hemisphere, the GOME data coverage after June 2003 is quite good because of the sufficient number of ground-based receiving stations that ensured data acquisition. On the other hand, in the SH the spatial coverage of the GOME measurements is quite sparse and limited to the areas covered by the German Antarctic Receiving Station O'Higgins (63° S, 54° W) operated by DLR and the ground station McMurdo (78° S, 167° E) operated by NASA. Thus, poleward of 60° S the number of grid cells for which GOME measurements are available is rather small. The last step for the computation of the adjustment for GOME is the interpolation of the monthly broad belt means to each 5°band. Figure 3 shows the final additive correction as a function of latitude and for each calendar month for the lowermost altitude layer. The offset varies from -2 DU (March in the tropics) to +3 DU (May and June in the NH poleward of 60°). In the tropics, the offset is negative during the entire year, whereas it varies between negative and positive values for the middle latitudes of both hemispheres and the southern high latitudes. In the high northern latitudes, the offset is positive throughout the year. Additionally, the black horizontal lines denote the annual average correction for each broadband latitude belt. In Fig. 2 the correction for GOME is denoted by the thick blue curve, which exhibits a seasonal variation around -1 DU.

After aligning the anomalies of GOME, SCIAMACHY, GOME-2A, and GOME-2B to the OMI anomalies, the merged anomalies $\delta_{\mathrm{merged}}$ are calculated as weighted averages of all $n$ instruments available for a particular spatio-temporal bin:

$$\delta_{\mathrm{merged}}(i,j,k,t) = \frac{\sum w_n(i,j,k,t) \cdot \delta_{n,\mathrm{ajd}}(i,j,k,t)}{\sum w_n(i,j,k,t)}. \tag{4}$$

$\delta_{n,\mathrm{ajd}}(i,j,k,t)$ denote the adjusted anomalies (as described above) in case of GOME, SCIAMACHY, GOME-2A, and GOME-2B and the original anomalies for OMI. The weights $w_n$ are the reciprocal squares of the corresponding uncertain-





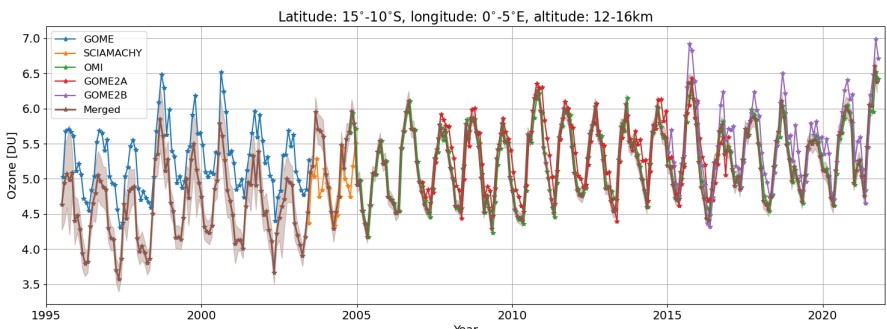

**Figure 4.** Ozone partial column in DU as a function of time for the $10$–$15°$ S, $0$–$5°$ E latitude-longitude bin and the altitude layer $12$–$16$ km: GOME (blue), SCIAMACHY (orange), OMI (green), GOME-2A (red), and GOME-2B (purple). The brown curve denotes the merged timeseries and the brown shading indicates the corresponding uncertainty.

ties: $w_n = 1/\sigma^2_{\delta_n}$ from Eq. 3. According to Taylor (2022), the uncertainties of the merged anomalies are estimated as

$$\sigma_{\mathrm{merged}}(i,j,k,t) = \frac{1}{\sqrt{\sum w_n(i,j,k,t)}}. \tag{5}$$

The last step in the merging procedure is to reconstruct absolute ozone values $\rho_{\mathrm{merged}}$ by adding back the seasonal cycle
from OMI to the merged anomalies

$$\rho_{\mathrm{merged}}(i,j,k,t) = \delta_{\mathrm{merged}}(i,j,k,t) + \rho_{m,\mathrm{OMI}}(i,j,k). \tag{6}$$

Figure 4 shows the final merged partial column and associated uncertainty as a function of time for the $10$–$15°$ S, $0$–$5°$ E latitude-longitude bin and the altitude layer $12$–$16$ km (brown curve). Additionally, the original partial columns for GOME (blue), SCIAMACHY (orange), OMI (green), GOME-2A (red), and GOME-2B (purple) are presented. Significant differences
between the original product and the merged product are observed for the GOME sensor from 1995 through 2004 due to the bias correction as described above (negative bias throughout the year for the tropics also for this altitude layer). Merged values from 2004 onward are dominated by the OMI measurements (green) as expected, due to the preceding alignment on the one hand, but also due the typically lower uncertainties and therefore higher weights for OMI on the other hand.

## 4 Homogenization with respect to GTO-ECV total columns

In principle the merged data record of ozone profiles generated as described in the previous section could be used as is for a possible assessment of the long-term evolution of ozone including trend estimation. However, the GTO-ECV total ozone data record (Sec. 2.3) with its well-proven quality, in particular as to the long-term stability (Garane et al., 2018), enables us to further improve the coherence of the merged profiles. An additional aim is to achieve consistency in terms of the total column ozone between both data records which are based on nearly the same satellite sensors.





The homogenization procedure which we propose largely takes advantage of the machine learning approach described in Xu et al. (2017) who developed an efficient method to retrieve ozone profile shapes from nadir-viewing satellite sensors. However, the main objective in this study is to establish an altitude-dependent scaling method that can be applied to the merged profiles created as described in Sec. 3 thereby aiming at matching the corresponding GTO-ECV total columns. The procedure mainly involves three steps including

1. the clustering of a subset of the merged profiles (see Sec. 4.1) aiming at assigning each profile of the subset to a class, i.e. a group of profiles of similar shape, and subsequently the classification of the remaining profiles, which are not used for the clustering (see Sec. 4.2);

    2. the calculation of profile Jacobians w.r.t. the total ozone columns per class from the previous step using a Neural Network (NN) approach (see Sec. 4.3). The Jacobians provide information about the altitude-dependent change of the partial
275        columns due to a change in the total column;

    3. and the altitude-dependent scaling of the merged profiles (see Sec. 4.4) in order to harmonize their integrated columns with the GTO-ECV total columns (per month and per $5° \times 5°$ grid cell).

The individual steps are described in detail in the following subsections.

## 4.1 Clustering algorithm

The main purpose of clustering the merged profiles is to generate a limited number of groups (preferably $\leq 20$) of ozone profiles from which the relation between changes in the total column and changes in the profile can be determined. Typically, ozone profiles are grouped on a latitudinally and/or monthly or seasonal basis, e.g. for building climatologies (e.g., McPeters and Labow, 2012; Sofieva et al., 2014). In these cases the number of groups (i.e. the number of spatio-temporal bins) would be much larger than 20. Moreover, information about the profile variability within a latitude belt or the interannual variability
would be lost. Stauffer et al. (2016, 2018) and Xu et al. (2017) have shown that clustering ozone profile data is also a reasonable way to describe and analyze their characteristics and their variability based on a considerably smaller number of groups.

The clustering of the data set of merged ozone profiles relies on measuring the degree of similarity between the individual profiles. As in Xu et al. (2017), we use the *k*-means clustering procedure (MacQueen, 1967) in order to find a certain number of separated subsets of profiles, which each contain profiles of similar shapes. The similarity of the profiles is assessed using
the Euclidean distance. The clustering procedure used here is based on a semi-supervised agglomerative hierarchical strategy. In addition to an unsupervised hierarchical clustering, this method involves extra background information into the process. For more details, we refer to Rokach (2010), Zheng and Li (2011), and Bair (2013). Based on two measures, the Silhouette coefficient (Rousseeuw, 1987) and the Davies-Bouldin index (Davies and Bouldin, 1979), Xu et al. (2017) have shown that the optimal number of ozone profile clusters is 11. We adopt this number for our study. Because the total number of ozone profiles
($\sim$650,000) in the merged data record would be too big for efficiently being used by the clustering procedure, a subset of 80,000 profiles is selected using the smart sampling technique (Loyola et al., 2016) and extracted in advance. These profiles serve as





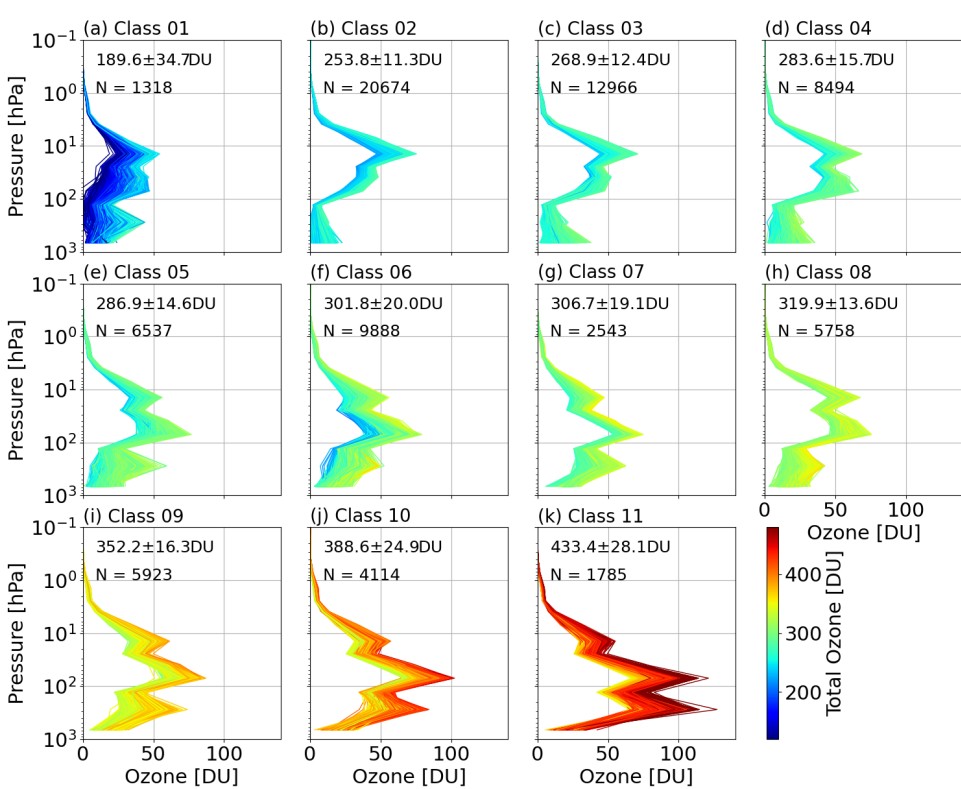

**Figure 5.** Set of 11 classes of ozone profiles, which is the outcome of the clustering procedure. The classes are ordered by the mean total column per class from 189.6 DU in panel (a) to 433.4 DU in panel (k). The mean total column and its $2\sigma$ standard deviation are provided in each panel in the top left corner. Additionally, the total number (N) of profiles in the respective class is given. The color assigned to each individual profile indicates the integrated total ozone column amount.

input for the clustering algorithm, and smart sampling ensures that this subset appropriately represents the characteristics of the entire record. The remaining ∼570,000 profiles, which are not part of the clustering, will be processed by a classification algorithm in a next step in order to assign them to one of the 11 classes (see Sec. 4.2). This approach allows for a relatively

simple future extension of the data record, because repeating both the clustering procedure and later on the calculation of profile Jacobians (see Sec. 4.3) can be avoided.

     Figure 5 shows the result of the clustering procedure, i.e. the 11 classes of ozone profiles of similar shapes. The classes are ordered by the mean total column per class from 189.6 DU in panel (a) to 433.4 DU in panel (k). The values provided in each panel in the top left corner denote the average ozone column in this class and its $2\sigma$ standard deviation. Additionally, the total

number (N) of profiles in the respective class is given. The color assigned to each individual profile indicates the integrated





total ozone column. The bulk of the profiles belongs to class 02 (Fig. 5(b), N=20,674), class 03 (Fig. 5(c), N=12,966), and class 04 (Fig. 5(d), N=8,494). Much less profiles belong to class 01 (Fig. 5(a), N=1318). The latter contains the profiles with minimum total column values. In addition to differences in total columns, the profiles shapes can be very different, e.g. with respect to the occurrence and location of the maxima, cf. classes 02 and 11. On the other hand, a few classes show apparent similarities. Classes 06 and 07 (see Fig. 5(f) and (g)) are similar with respect to the mean total ozone column (301.8 and 306.7 DU, respectively) and also the profile shape.

For each class we analyze its composition with respect to the distribution of the total columns as well as the month and latitude from which the profiles come in more detail. Figure 6 shows this statistics exemplarily for class 11 ((a)-(c)), class 02 ((d)-(f)), and class 01 ((g)-(i)), respectively. The distribution of total columns is shown in the left panels ((a), (d), (g)), the distribution of the months is shown in the middle panels ((b), (e), (h)), and the distribution of the latitudes is shown in the right panels ((c), (f), (i)).

The profiles assigned to class 11 have large total columns of 433.4±28.1 DU (Fig. 5(k); they are mostly located between 45° N and 90° N, and the distribution of the months has its maximum from February to April (boreal late winter and spring). On the other hand, profiles assigned to class 02 are characterized by much less total ozone covering only a small range from 240 to 280 DU (Fig. 6 (d)). In general all months are covered equally with only a slight maximum of the distribution from December to March and a minimum in June and July. Almost all profiles in class 02 are located in the Tropics between 30° S and 30° N with a maximum from 15° S to 15° N. The profiles of class 01 (g)-(h) can be clearly assigned to the ozone hole season from September to November in the high southern latitudes poleward of 60° S with low average ozone columns around 175 DU.

## 4.2 Classification of remaining profiles

As described in Sec. 4.1, for the clustering of the profiles and the calculation of the derivatives (Sec. 4.3) a subset of 80,000 profiles selected from the complete merged data record (Sec. 3) was used. During the clustering procedure each of these profiles was assigned to one of the 11 classes. However, the altitude-dependent scaling of the complete data record requires, that, in addition to the 80,000 profiles, each of the remaining profiles, which were not used for the clustering procedure, is assigned to a certain class as well, in order to select the appropriate set of derivatives for the scaling. Therefore, we apply the $k$-nearest neighbors classification algorithm (Cover and Hart, 1967) to the remaining profiles to determine the most appropriate class membership. In this non-parametric approach the profiles are classified by a plurality vote of their neighbors, and the profiles are assigned to that class most similar to the $k$ nearest neighbors. As training data we use the set of 11 classes of ozone profiles as generated in Sec. 4.1. In our case $k = 2$, and the points are weighted by the inverse of their distance, i.e. nearer neighbors contribute to a greater extent.

The mean accuracy of this method has been determined to about 98.5% based on test data for which the correct class was known. Finally, each of the remaining profiles is assigned to one of the 11 classes. We analyze the distribution of the profiles in the classes (i.e. the percentage of profiles assigned to a respective class) and compare the distribution as it comes out from the clustering with the distribution as outcome of the classification. Both distributions are almost identical and differ only by



**Figure 6.** Composition of class 11 ((a)-(c)), class 02 ((d)-(f)), and class 01 ((g)-(i)) with respect to the distribution of the total column ((a), (d), (g)), the month from which the profiles come ((b), (e), (h)), and the respective latitude ((c), (f), (i)). In panels (a), (d), and (g), total column amounts are additionally highlighted by the colors. In panels (b), (e), and (h), blue, green, yellow, and red denote profiles from December to February, March to May, June to August, and September to November, respectively. In panels (c), (f), (i), latitude belts (width of 15°) are additionally highlighted by colors from dark blue (90°–60° S) to dark red (60°–90° N).





340  up to 0.4%. For example, 31.0% of the profiles from the subset were assigned to class 02 during the clustering (see Fig. 5(b)) and 31.4% of the remaining profiles were assigned to this class as outcome of the classification. For class 11 these values are 3.0% and 3.1%, respectively.

Figure 7 shows global maps of ozone profile classes assigned to the monthly mean merged profiles for April 2014 (a) and October 2014 (b). Almost the entire tropical band (25°S–25°N) contains profile shapes assigned to class 02 (light yellow) with 345  mean total ozone amounts of ∼254DU. As expected, in April in the high latitudes of the Northern Hemisphere profiles with maximum ozone amounts above 400 DU (yellow) are found. On the other hand, profiles with extremely low ozone amounts (class 01, TOC<200 DU) are found in high southern latitudes in October (turquoise). The adjacent classes toward the equator are class 04 (red) and class 06 (orange), which are characterized by much higher ozone columns, but completely different shapes as for example classes 02 and 03. The results agree well with Figure 6. In general, the zonal variability of the classes is 350  small with a few exceptions in particular poleward of 40° in both hemispheres during the winter months (not shown).

### 4.3   Calculation of ozon profile Jacobians using Neural Networks

One of the main objectives of this study — in addition to merging profiles from the individual sensors — is the development of an altitude-dependent scaling approach that can be applied to the merged profiles from Sec. 3 in order to match the total ozone columns provided with the GTO-ECV data record. For this purpose we use the standard feed-forward neural network 355  algorithm implementation developed by Molina García (2022), which offers the possibility to extract derivatives with respect to the input variables, e.g., total ozone. These derivatives are obtained by automatic differentiation (Molina García et al., 2018), and they provide information about the altitude-dependent change of the profile as a result of a change in the total column. For each of the 11 classes compiled in Sec. 4.1 a separate neural network is trained based on the samples shown in Fig. 5. The next step is to define the optimum configuration for the neural networks which comprise an input layer, one or more hidden layers, 360  and an output layer. A performance study based on 420 different settings as listed in Table 2 was carried out. For the input layer, different combinations of the parameters total ozone (selected for each configuration), month, latitude, and longitude were used. The output layer then contains the 19 partial columns of the profile. The performance of each NN configuration was evaluated based on external validation data which were extracted from the sample not used in the training. The main findings of this analysis are as follows: Compared to an NN configuration which uses only total ozone in the input layer, configurations 365  with additional input parameters significantly improve the retrieval result for the 19 partial columns. Best agreement was found for 3 (total ozone, latitude, and month) and 4 (total ozone, latitude, longitude, and month) input parameters. On the other hand, varying the number of hidden layers and the number of neurons per hidden layer impacts the quality to a lesser extent.

For our purpose, i.e. the development of an altitude-dependent scaling, it is needed to obtain a quite robust estimate of the derivatives w.r.t. the total columns. This can be achieved by compiling an ensemble of NNs (Loyola, 2006) each of which 370  providing derivatives that can be extracted. Finally the ensemble's median derivative will be used for the scaling task which is described in the next Section. Based on the results of the performance study conducted in advance the ensemble of NNs consists of the following configurations (see also Table 3): in the input layer 3 and 4 neurons are used, the number of hidden layers is set to 2, and the number of neurons in each hidden layer is varied from 10 to 30 in steps of 2. The total number of





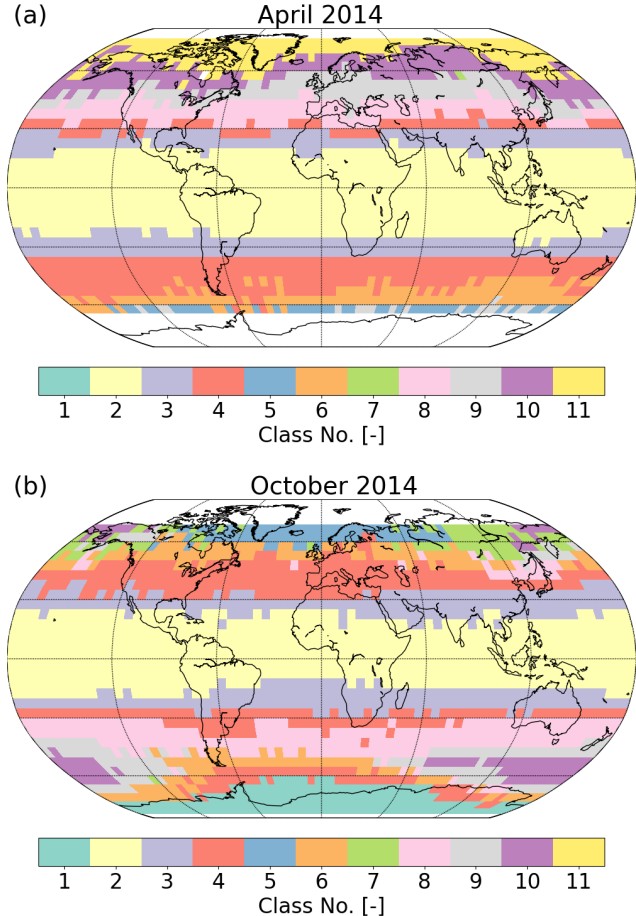

**Figure 7.** Global map of ozone profile classes assigned to the monthly mean merged profiles for (a) April 2014 and (b) October 2014. White grid cells denote that no data is available, mainly due to polar night conditions.

**Table 2.** Overview of 420 neural network configurations for the performance study to find appropriate settings.

| Parameter | Values |
|---|---|
| Number and [list of input neurons] | 1 [TOC], 2 [TOC, month], 2 [TOC, latitude], |
| | 3 [TOC, latitude, month], 4 [TOC, latitude, longitude, month] |
| Number of hidden layers | 1, 2, 3 |
| Number of neurons per hidden layer | 10, 20, 30, 50 |
| Output | 19 partial columns for profile layers 1–19 (surface to 80km) |

members in the ensemble of NNs is 242. In contrast to the performance study, the number of output neurons is reduced to



**Table 3.** Overview of final neural network configurations for generating an ensemble of 242 NNs for each of the 11 classes for the calculation of derivatives.

| Parameter | Values |
| --- | --- |
| Number and [list of input neurons] | 3 [TOC, latitude, month], 4 [TOC, latitude, longitude, month] |
| Number of hidden layers | 2 |
| Number of neurons per hidden layer | 10, ..., 30 in steps of 2 |
| Output | 14 partial columns for profile layers 1–14 (surface to ∼64km), layers 15–19 are omitted |

14, which means that only the 14 lowermost profile layers from the surface to ∼64 km are used. Since the amount of ozone above 64 km is only about 0.02% of the total column, we decided to omit these layers from the scaling so that they will remain unchanged while scaling the other layers.

For each class each ensemble of derivatives extracted from the trained NNs is further divided into 10 subgroups with respect to the total ozone column. Figure 8 shows the ensemble of 242 derivatives as a function of altitude for class 01 (Fig. 8(a)),
class 02 (Fig. 8(b)), and class 11 (Fig. 8(c)). The thick black line denotes the median derivative. Each blue and orange curve correspond to a single neural network configuration with either 3 (blue) or 4 (orange) input neurons and different configurations of neurons in the two hidden layers (see Table 3). As for the ozone profiles themselves (Fig. 5) the shapes of the derivatives can be quite different. Maxima and minima are located at different altitudes for the different classes. Derivatives for class 01, which contains mainly profiles from the high southern latitudes from September to November (i.e. the ozone hole season),
are maximum between 100 and 30 hPa (∼16–24 km). The change in these layers is about 0.22 DU/DU. In the lowermost troposphere the derivative is small. On the other hand, derivatives of class 02 (Figure 8(b)) have their maximum in this layer from 0–5.5 km, a minimum near the tropopause, and another maximum at ∼30 km (20–10 hPa). The derivatives for class 11 have a strong single maximum of 0.35 DU/DU in the upper troposphere (6–12 km).

### 4.4 Scaling the merged ozone profiles

The last step in the homogenization procedure is the scaling of the merged profiles from Sec. 3 aiming at matching the total ozone columns provided with the GTO-ECV data record (Sec. 2.3). As input for the scaling of each individual merged ozone profile $\rho_{\mathrm{merged}}(i,j,k,t)$ for latitude bin $i$, longitude bin $j$, altitude layer $k$, and month $t$ we need

  1. the information to which class the profile was assigned to (see Sec. 4.2);

  2. the derivatives corresponding to this respective class (see Sec. 4.3);

3. both the total ozone column provided with the GTO-ECV data record ($\mathrm{TOC_{GTO}}$) and the integrated column of the merged profile ($\mathrm{TOC_{Profile}}$) from which the difference $\Delta\mathrm{TOC}$ between both data records as $\Delta\mathrm{TOC}(i,j,t) = \mathrm{TOC_{GTO}}(i,j,t) - \mathrm{TOC_{Profile}}(i,j,t)$ is computed.





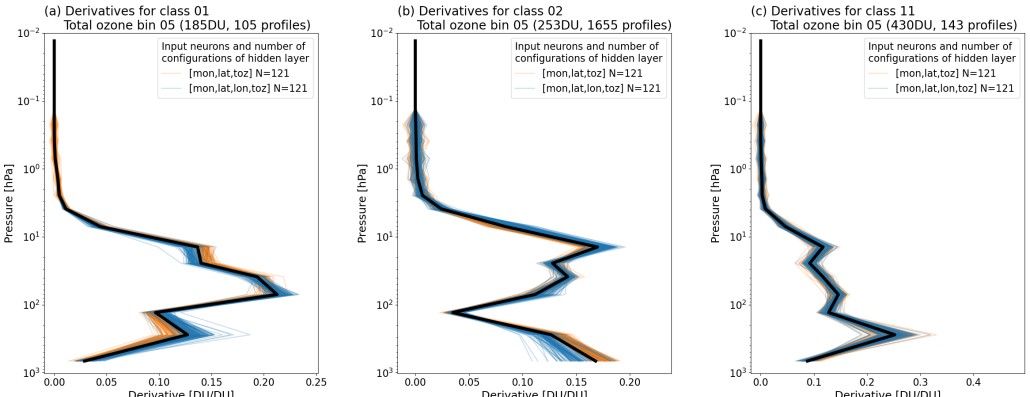

**Figure 8.** Ensemble of derivatives w.r.t. total ozone as a function of altitude obtained from the NN training for class 01 (a), class 02 (b), and class 11 (c). The thick black lines denote the ensemble's median derivative. Blue and orange curves correspond to single network configurations with either 3 (blue) or 4 (orange) input neurons and different configurations of neurons in the two hidden layers (see also Table 3). For each class the complete ensemble consists of 242 members.

The total ozone difference $\Delta$TOC between the GTO-ECV data record and the merged profile record is exemplarily shown in Fig. 9 for January and July 2014. The global mean difference for these two months is $3.7 \pm 4.9$ DU and $0.9 \pm 5.1$ DU,
respectively. In general, the deviations and their spatial variability are smaller in the tropics and the middle latitudes of the summer hemispheres, whereas they significantly increase towards the middle latitudes of the winter hemispheres. In January 2014, the difference is mostly positive for nearly the entire globe and negative deviations are found only for example poleward of $50°$ N for $120°$–$180°$ E. We do not see apparent differences between land and ocean surfaces. On the other hand, the pattern in July 2014 indicates more negative deviations over land (e.g., southern Africa, the Arabian Peninsula, and southeast China)
than over water (e.g., eastern and northern Pacific and the northern Atlantic). In July 2014 in the middle latitudes of the southern hemisphere, the differences are positive in the band $30°$–$45°$ S and negative for $45°$–$60°$ S, in particular in the region $45°$–$150°$ E.

The total ozone difference is finally used to scale the merged profiles following

$$\rho_{\text{merged,scaled}}(i,j,k,t) = \rho_{\text{merged}}(i,j,k,t) + \Delta\text{TOC}(i,j,t) \cdot \frac{\partial\rho(k)}{\partial\text{TOC}}, \qquad (7)$$

where $\frac{\partial\rho(k)}{\partial\text{TOC}}$ denotes the derivative as a function of altitude with respect to the total ozone column obtained from the neural network ensemble as described in Sec. 4.3.

Figure 10 shows the impact of the altitude-dependent scaling of the profiles for January ((a) and (b)) and July ((c) and (d)) 2014 for the tropics ($30°$ S–$30°$ N, orange) and the middle latitudes of the winter hemispheres (blue), i.e. $30°$–$60°$ N for January 2014 and $30°$–$60°$ S for July 2014, respectively. Figures 10 (a) and (c) show the mean profiles in that latitude belts. The solid
curves denote the scaled profiles and the shading indicates the standard deviation in that latitude band. The dotted curves and the errorbars denote the merged profiles before the scaling is applied. As expected, the profile shapes for the tropics and the





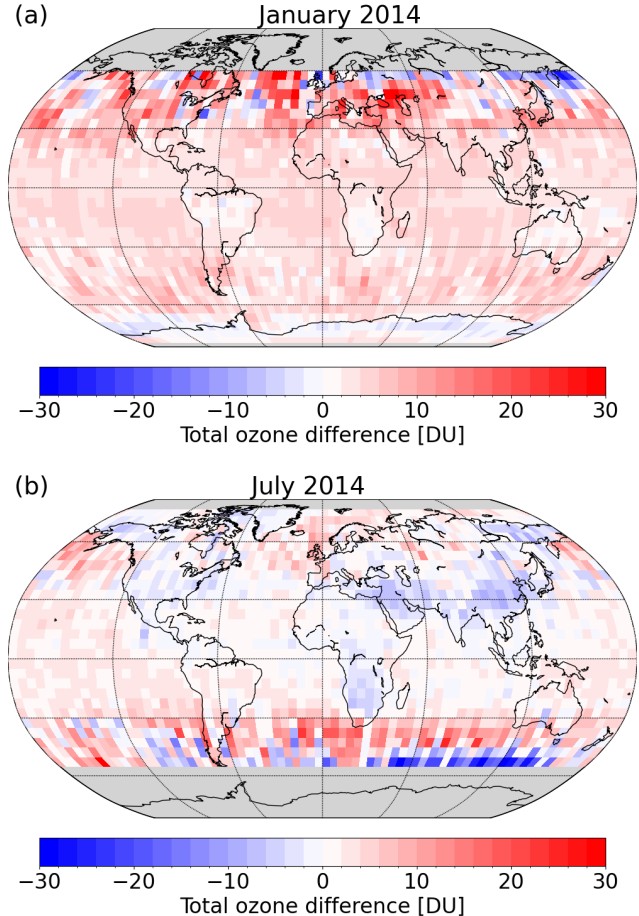

**Figure 9.** Total ozone difference [DU] between the GTO-ECV data record and the merged profile record for January (a) and July (b) 2014. Grey shaded grid cells denote that no data is available, mainly due to polar night conditions.

middle latitudes are quite different. In the low latitudes the maximum of the profile is located at higher altitudes and the total column is smaller than in the middle latitudes. The difference between the scaled and the original profiles (solid vs. dotted curves) is rather small. Figures 10 (b) and (d) show the corresponding mean difference between the merged profiles with and

without the scaling applied for the same latitude bands. As can be anticipated from Fig. 9, in general, the mean deviations are positive for both the tropics and the middle latitudes and also for both months. The differences are larger in the middle latitudes ($\sim 0.5 - 1.0\,\text{DU}$) compared to the tropics ($0 - 0.5\,\text{DU}$). In the middle latitudes the variability is larger, too, as was expected from the pattern of the total ozone differences (see Fig. 9). Largest changes ($-2 - +4\,\text{DU}$) can occur at $\sim 300\,\text{hPa}$.

In order to evaluate the impact of the scaling w.r.t. GTO-ECV over the entire period of the profile record, Fig.,11 shows

the ratio of ozone from the scaled profiles to ozone from the profiles before the scaling from 1995 through 2021 for the three latitude bands 35°–50°N (a), 20°S–20°N (b), and 35°–50°S (c). A 12-month running mean was applied to the ratios that are





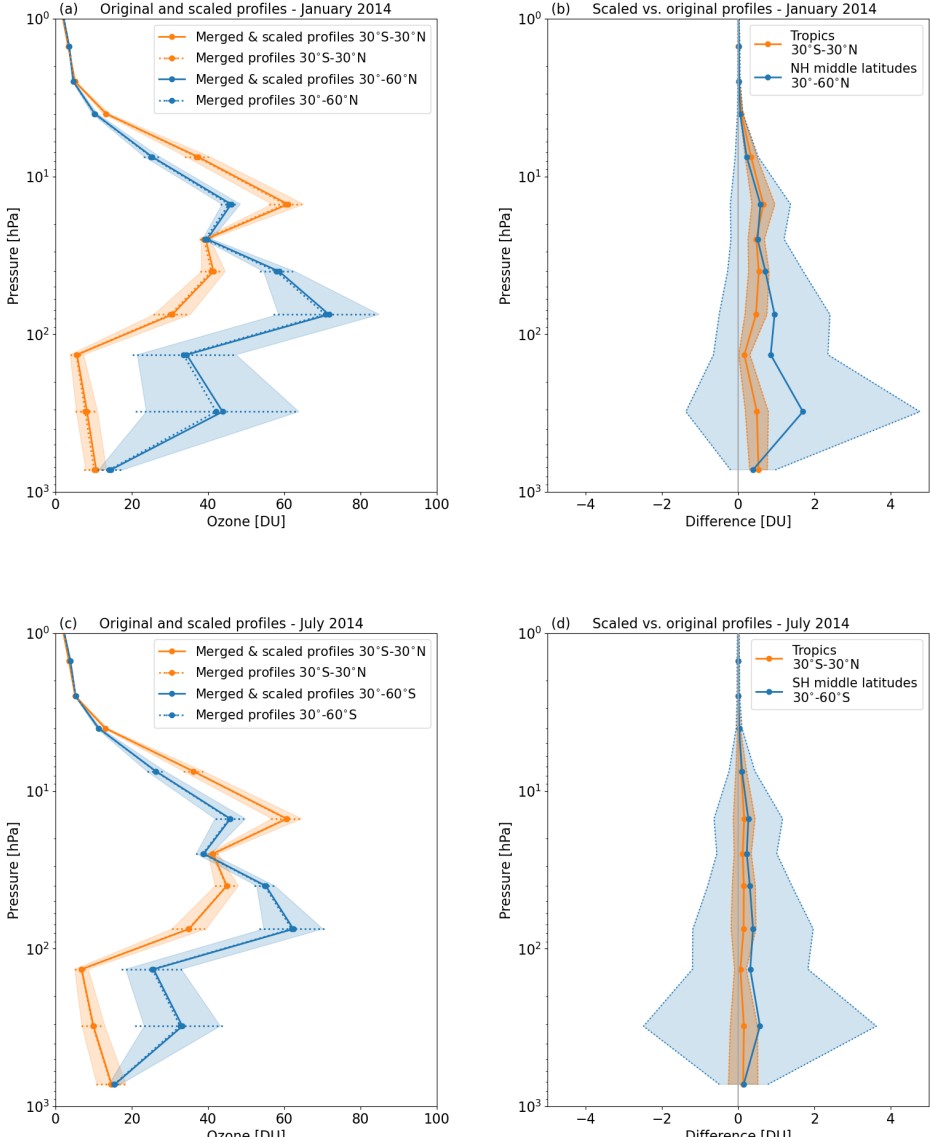

**Figure 10.** Mean ozone profiles for January 2014 (a) and July 2014 (c) for two latitude bands: 30° S–30° N (orange) and 30°-60° N (blue) for January (a) and 30° S–30° N (orange) and 30°-60° S (blue) for July (c). Solid curves denote the scaled profiles and the shading indicates the standard deviation. Dotted curves and the errorbars denote the merged profiles before the scaling is applied. Panels (b) and (d) show the corresponding mean difference between the merged profiles with and without the scaling applied for 30° S–30° N (orange) and 30°-60° N (blue) for January (b) and 30° S–30° N (orange) and 30°-60° S (blue) for July (d).





given for the total ozone column (thick blue curve), and the partial ozone columns in three layers surface–450 hPa (0–6 km, red), 100–50 hPa (16–20 km, orange), and 3-2 hPa (40–43 km, green), respectively. Additionally, the grey horizontal bars in panel (b) indicate the period for each sensor included in the merged product.

The mean ratio for the total column (thick blue curves) is about 1.01-1.02 for all three latitude bands. The scaling of the partial columns in the 100–50 hPa layer (orange curves) closely follows the scaling of the total columns for all belts. The scaling in the uppermost layer (3–2 hPa, green curves) is very close to one. The largest deviations of up to 10% between the scaled and the non-scaled time series and an apparent variation with time are seen for the lowermost tropospheric layer surface–450 hPa (red curves). This is in agreement with Fig. 10 (b) and (d). The ratios reach maximum values of ∼4% in the NH, ∼8% in the

SH, and up to 12% for the tropical band. The variability with time is somewhat larger during the first decade from 1995 to 2004, when the product solely consists of GOME measurements from 1995–2002 and of GOME and SCIAMACHY data from 2003–2004. In the SH, the scaling factors indicate a quite abrupt decrease end of 2004, when OMI was added to GOP-ECV and GOME and SCIAMACHY are no longer included. From 2007 to 2015 the merged product consists of OMI and GOME-2A measurements, and during that period the ratio decreases from maximum values to minimum values in each latitude band. In

the tropical belt, the scaling factors significantly decrease from about 10% to about 3%. The transition from GOME-2A to GOME-2B (from 2015 onward) then again leads to increasing ratios until 2019.

## 5   Climatological ozone distributions

In this section, we present examples of the climatological ozone distribution from 1995–2021 derived from the final merged and scaled ozone profile record. We selected two months (April and October) and show the global spatial distribution of the

integrated column amount and for two selected layers (i) surface–450 hPa (0–6 km) and (ii) 100–50 hPa (16–20 km) in Fig. 12. In Fig. 12, the total column ozone is presented for April (a) and October (b), and it indicates the expected large-scale pattern. In April, maximum ozone columns above 400 DU occur in the middle and high latitudes of the NH, whereas ozone in October reaches minimum values below 200 DU poleward of 60°S. In the tropical region, ozone amounts show little variation and are about 250 DU in both months. As can be expected from the scaling, this distribution agrees well with the distribution obtained

from the GTO-ECV data record (see e.g., Coldewey-Egbers et al., 2020, their Fig. 4).

Fig. 12 (c) and (d) show the partial column amounts in April and October for the lowermost tropospheric layer from the surface to 450 hPa. In the Tropics, minimum values are found over the Pacific, whereas the distribution has a year-round maximum in the South Atlantic region. This maximum is most pronounced in September (not shown) and October and this seasonality is due to large-scale transport of ozone and a seasonal variability in ozone production sources (e.g., biomass burning

or lightning, Ziemke et al., 2011). Another maximum in lower tropospheric ozone occurs over China, Japan, and India which are prominent regions for significant emissions of ozone precursors, e.g., $NO_x$ (Elshorbany et al., 2024). The overall pattern of ozone in this layer agrees quite well with the results shown in Pope et al. (2023). Their data record is also based on RAL ozone profile products, but uses a different set of sensors and is limited to the lower tropospheric column from 1996–2017.





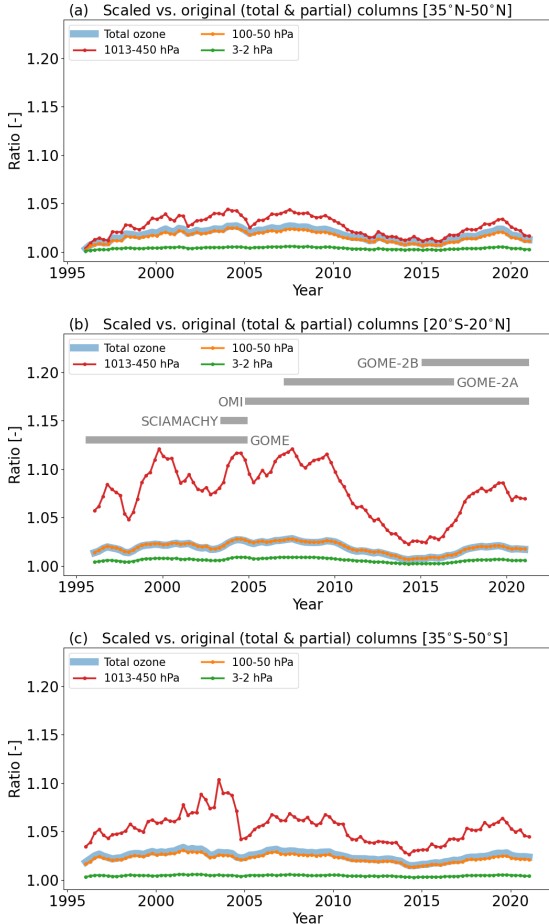

**Figure 11.** Ratio of scaled and original ozone as a function of time (1995–2021) for three latitude bands: (a) 35°–50°N, (b) 20°S–20°N, and (c) 35°–50°S. A 12-month running mean was applied. The ratio is shown for the total ozone column (thick blue curve), and the partial ozone columns in the layers surface–450 hPa (red), 100–50 hPa (orange), and 3-2 hPa (green). Additionally, the grey horizontal bars in panel (b) indicate the period for each sensor included in the merged product.

The distribution for the layer 100–50 hPa (Fig. 12 (e)-(f)) can be compared qualitatively with the examples presented by
Sofieva et al. (2021, their Fig. 9). In April, the distribution is quite homogeneous in longitude, and maximum values occur in the NH middle and high latitudes. On the other hand, in October, longitudinal structures in the SH come along with the polar vortex. In a subsequent study a more detailed and qualitative comparison with other data records will be performed.

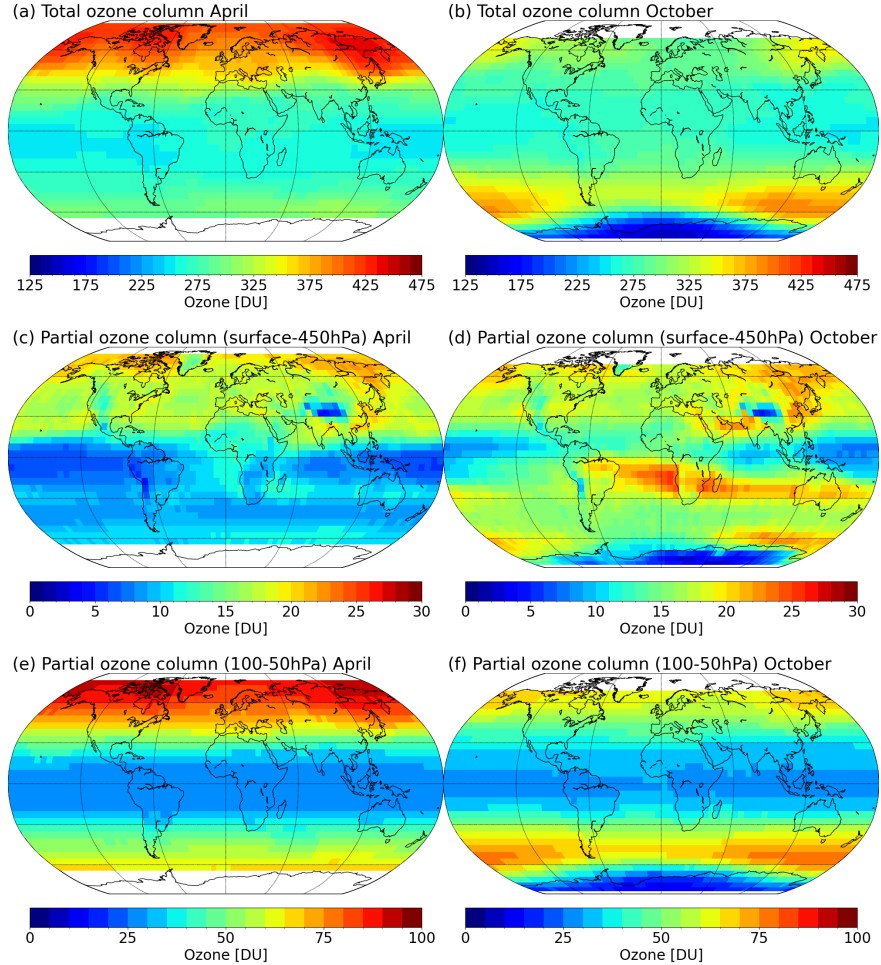

**Figure 12.** Climatological ozone distribution derived from GOP-ECV 1995-2021 for the integrated total column ((a) and (b)), and the partial column amounts in the layers: surface-450 hPa (0–6 km, (c) and (d)), and 100–50 hPa (16–20 km, (e) and (f)). For the total column and each layer the values for April ((a), (c), (e)) and October ((b), (d), (f)) are shown. White grid cells denote that no data is available, mainly due to polar night conditions.

## 6 Summary and outlook

In this paper, we introduce the new GOP-ECV climate data record of ozone profiles developed by DLR in the framework of the ESA-CCI+ ozone project. It is a compilation of measurements from five nadir-viewing UVN satellite sensors including GOME/ERS-2, SCIAMACHY/ENVISAT, OMI/Aura, GOME-2/MetOp-A, and GOME-2/MetOp-B, which are merged into a single coherent time series. GOP-ECV covers the 26-year period from July 1995 through October 2021, and it provides monthly mean ozone profiles at a spatial resolution of $5° \times 5°$ latitude by longitude. The profiles are given as partial columns



for 19 atmospheric layers ranging from the surface up to 80 km. The underlying profile retrieval algorithm is the RAL scheme,
which has sensitivity to tropospheric as well as stratospheric ozone.

Profiles from the individual instruments are first homogenized, thereby taking into account inter-sensor biases and drifts, and
then merged into a combined record. In a next step, the merged time series is further harmonized with the GOME-type Total
Ozone Essential Climate Variable (GTO-ECV) data record, which is based on nearly the same satellite sensors. GTO-ECV
possesses an excellent long-term stability and with the homogenization an improvement of the coherence and the stability
of the merged profiles can be achieved. For this purpose, an altitude-dependent scaling has been developed that makes use
of ozone profile Jacobians obtained from a Machine Learning approach. The scaling finally ensures a harmonization of both
GTO-ECV and GOP-ECV in terms of the total column.

Climatological ozone distributions derived from the final GOP-ECV data for selected atmospheric layers agree with spatial
patterns obtained from other long-term data records. In a subsequent study, detailed results of the geophysical validation
using ground-based data and results of more systematic and quantitative comparisons with other satellite-based records will
be presented. A special emphasis will be put on the investigation of the variability and long-term changes of tropospheric
ozone. As a first application of GOP-ECV, the lowermost tropospheric profile layer (surface–450 hPa) contributed to a study
performed by Keppens et al. (2024), that aimed at the harmonization of various tropospheric ozone data records from satellites.

For the near future, it is planned to extend GOP-ECV both in time and with observations from two additional satellite sensors,
viz., the TROPOpheric Monitoring Instrument (TROPOMI) onboard the Sentinel-5 Precursor (from May 2018 onward) and
GOME-2 onboard Metop-C (from January 2019 onward). Moreover, for the sensors GOME, OMI, GOME-2A, and GOME-2B
reprocessed full mission Level-2 profile data based on an improved RAL retrieval scheme will be included. GTO-ECV as a
baseline for the profile scaling will be updated and extended as well. We will take advantage of reprocessed full mission OMI
total ozone data retrieved with GODFIT using as input the new OMI Level-1 Collection 4 (Kleipool et al., 2022).

*Data availability.* The merged GOP-ECV ozone profile product is developed within ESA's Climate Change Initiative (CCI+) on ozone and
is available through the Ozone-CCI+ website https://climate.esa.int/en/projects/ozone/ (last access: 07 December 2024). The nadir profile
Level-3 data and the GTO-ECV product can be downloaded from C3S https://cds.climate.copernicus.eu/datasets/satellite-ozone-v1?tab=
download (last access: 07 December 2024, Coldewey-Egbers and Loyola). The L2 data sets produced by the RAL scheme are to be archived
for public access on CEDA https://www.ceda.ac.uk/ (last access: 09 December 2024).

*Author contributions.* MCE performed the analysis of the satellite data and the generation of the data record. DL initiated this work and
provided scientific advice to the design of the data record. BL, RS, and BK provided the Level-2 data and supported the analysis. DH, MVR,
and CR contributed to the discussion. MCE prepared the major part of the manuscript with input from all co-authors.



*Competing interests.* At least one of the (co-)authors is a member of the editorial board of Atmospheric Measurement Techniques. The authors have no other competing interests to declare.

500 *Acknowledgements.* RAL's scheme to retrieve ozone height resolved data from satellite UV sounders was funded through UK's National Centre for Earth Observation and ESA's Climate Change Initiative. Production of L2 data sets was funded also through EU's Copernicus Climate Change Services (C3S) - Atmospheric Composition.

*Financial support.* This research has been supported by the European Space Agency CCI+ ozone project (grant 4000126562/19/I-NB). The article processing charges for this open-access publication were covered by the German Aerospace Center (DLR).



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
