# Peer review of "The novel GOME-type Ozone Profile Essential Climate Variable (GOP-ECV) data record covering the past 26 years"

_Atmospheric Measurement Techniques, 2024_

## Referee Comment (RC1)

The manuscript by Coldewey-Egbers et al. presents a very interesting and innovative merged ozone profile data set from nadir observations (GOP-ECV). The authors introduce the main aim of the work, the harmonization procedure used for the merging of the data sets and the scaling procedure with respect to the GTO-ECV time series. This paper fits the scope of AMT, it is well written and scientifically sound. I found the given explanations overall convincing, with clear descriptions of the multiple, and sometimes complicated, steps. From my side, I only have some minor comments on specific aspects and some technical corrections.

**Specific comments/questions**

- I am wondering about the usage of SCIAMACHY data set over the period 2002-2004, as reported in Tab.1. Is it only used over these two years for the generation of GOP-ECV? Does the usage until 2012 have a negative impact on the merged dataset? I would suggest to include a short explanation in the manuscript about this choice. Regarding limb observations, in Sofieva et al. (2017) the usage of the first months of SCIAMACHY were not recommended due to some unexplained features in the anomalies (for SAGE-CCI-OMPS it is used from August 2003). Have you noticed any larger discrepancy in nadir data at the beginning of the SCIAMACHY period?

- OMI time series is used in this work as reference for the other datasets, also to remove drifts. Does the drift affecting OMI total column time series or its row-anomaly, e.g. Torres et al. (2018), Gaudel et al. (2024) supplements, have any potential impact on this choice?

- Just a couple of clarifications regarding the used neural network approach, as I am not very familiar with this. Is the described NN approach a sort of ozone profile retrieval? Are the derivatives extracted in a second step or directly provided by the NN?

  In the simplest case from Tab.2, are you feeding the NN only with TOC for each class and let the hidden layers find a mapping between TOC and profile shape? It seems to me that in this case there could be profiles with the same TOC but different shape even within the same class. I just wonder how the NN is able to distribute the TOC variations vertically without having unique solutions.

  How do you get to the number 420 in Table 2? I understand the 242 possible combinations in Table 3, as you have 2 hidden layers and 11 possibilities for each, times 2 options for the inputs, but I could not get to 420 combinations in Tab.1.

- A side note: is the spiky shape of the profiles in Fig. 5 a feature of the RAL retrievals? Most of them tend to have three local maxima.

**Technical corrections**

Line 13: I would add "presented in this manuscript" after "the homogenization".

Line24: "banned" → "prevented"

Line 30-31: "the middle latitudes of the Northern Hemisphere" → "at northern mid-latitudes"

Line 59: Add a , after "data sets".

Lines ~60: You could mention the advantage/disadvantage to use limb or nadir data to retrieve profiles in terms of vertical and spatial resolution.

Line 79: "allows us to generate" → "enables the generation of"

Line 80: "in particular important" → "particularly important"

Line 82: What is it meant with "investigation of changes in the profile"? Stratospheric ozone trends?

Line 85: "enables us to assess" → "facilitates the assessment of"

Line 97: Add , after "ozone profiles".

Line 98: The UVN acronym was already introduced in the previous page.

Line 160: Add , after "level-2 products".

Line 202: Also at northern mid-latitudes SCIAMACHY has a positive bias.

Lines 205-207: I would move this last three lines to the beginning of the paragraph (line 194), as these are general considerations about the seasonal cycle.

Line 216: "drift" is repeated two times.

Lines 220-222: Do you plot in Fig. 2 the fit, for example, to (GOME-OMI) anomalies (as you state in the text) or to OMI-GOME?

Line 227-229: I find hard to read the sentence starting with "From these deviations…". I suggest to re-formulate such as: "From the time series of the offsets in each available spatial bin, at first, we calculate averages for each calendar month ("climatologies") and then we average them over five broad latitude bands…".

Line 242: "aligning" or "harmonizing"?

Line 262: "in particular as to the…" → "in particular in terms of the..."

Line 328: Remove , after "requires".

Line 360: "of the parameters total ozone…" → I would add "of the parameters, i.e. total ozone…"

Line 371: Add , after "in advance"

Line 402: "only for example poleward of 50° N for 120°-180°" → "mostly at latitudes poleward of 50° N and at 120°-180° E."

Line 436: "measurements from" → "measurements over"; "data from" → "data over".

I would remove Line 446 as it repeats what said in the previous lines.

---

## Author Comment (AC1)

**Reply to Anonymous Referee #1**

We thank anonymous referee #1 for her/his positive review and the helpful comments and corrections. Please find below the reviewer's comments (in black), our responses (in blue), and changes or additions to the text (in red).

**Anonymous Referee #1, 06 Mar 2025**

The manuscript by Coldewey-Egbers et al. presents a very interesting and innovative merged ozone profile data set from nadir observations (GOP-ECV). The authors introduce the main aim of the work, the harmonization procedure used for the merging of the data sets and the scaling procedure with respect to the GTO-ECV time series. This paper fits the scope of AMT, it is well written and scientifically sound. I found the given explanations overall convincing, with clear descriptions of the multiple, and sometimes complicated, steps. From my side, I only have some minor comments on specific aspects and some technical corrections.

**Specific comments/questions**

I am wondering about the usage of SCIAMACHY data set over the period 2002-2004, as reported in Tab.1. Is it only used over these two years for the generation of GOP-ECV? Does the usage until 2012 have a negative impact on the merged dataset? I would suggest to include a short explanation in the manuscript about this choice. Regarding limb observations, in Sofieva et al. (2017) the usage of the first months of SCIAMACHY were not recommended due to some unexplained features in the anomalies (for SAGE-CCI-OMPS it is used from August 2003). Have you noticed any larger discrepancy in nadir data at the beginning of the SCIAMACHY period?

 $\rightarrow$  SCIAMACHY data is used for the generation of GOP-ECV as follows: We use 6 years (2005-2010) of the overlap period with OMI for the estimation of the bias and drift correction, that is then applied to the entire SCIAMACHY time period (incl. 2002-2004). Since we do not have overlap with OMI from 2002-2004, we cannot compare our results with the findings in Sofieva et al., 2017.

In the final merged data record, we use SCIAMACHY data only for the period 2002-2004 in order to bridge the gap between GOME (this sensor unfortunately lost its global coverage in June 2003) and the beginning of the OMI measurements. OMI then provides a very good spatial coverage. We included a short explanation in Section 3.

OMI time series is used in this work as reference for the other datasets, also to remove drifts. Does the drift affecting OMI total column time series or its row-anomaly, e.g. Torres et al. (2018), Gaudel et al. (2024) supplements, have any potential impact on this choice?

 $\rightarrow$  We agree that the quality of the merged product will be determined to a large extent by the quality and long-term stability of the OMI data record. OMI is used not only a a reference sensor for the profile product, but also as a reference sensor for the total column product GTO-ECV. It is possible that the changing cross-track coverage during the mission could influence the OMI time series if sampling of L2 data is not restricted to the cross-track pixels which were continuously usable.

In the framework of the EU Copernicus Climate Change Service (C3S) ozone project, OMI total ozone and ozone profile data are validated using ground-based data on a regular basis. For both data records, it was found that the drift with respect to the ground reference is mostly insignificant (see C3S PQAR).

C3S Product Quality Assessment Report: https://dast.copernicus-climate.eu/documents/satelliteozone/C3S2\_312b\_Lot2\_2024/C3S2\_312a\_Lot2\_D-WP2\_FDDP-PQAR\_202311\_O3\_v3.3\_final.pdf Just a couple of clarifications regarding the used neural network approach, as I am not very familiar with this.

Is the described NN approach a sort of ozone profile retrieval?

 $\rightarrow$  In our study, the sole purpose of the NN approach is the estimation of the Jacobians, which provide the information about the altitude-dependent change of the profile due to a change in the total ozone column. During the training process, the ozone profile serves as the output, but the mapping from the input (total ozone, month, lat, lon) to the profile (output) is not used for the construction of the GOP-ECV data record.

However, in a study by Xu et al. (2017), a similar NN approach is used as part of an ozone profile shape retrieval, i.e. for the scaling of the profile shape according to the total column (see their Fig. 1(b) and Sec. II-E).

J. Xu, O. Schüssler, D. G. L. Rodriguez, F. Romahn and A. Doicu, "A Novel Ozone Profile Shape Retrieval Using Full-Physics Inverse Learning Machine (FP-ILM)," in *IEEE Journal of Selected Topics in Applied Earth Observations and Remote Sensing*, Vol. 10, No. 12, pp. 5442-5457, doi: 10.1109/JSTARS.2017.2740168, 2017.

Are the derivatives extracted in a second step or directly provided by the NN?

→ The derivatives are directly provided by this NN implementation developed by Molina García:

Molina García, V.: Retrieval of cloud properties from EPIC/DSCOVR, Ph.D. thesis, Technical University of Munich, https://elib.dlr.de/194303/1/MolinaGarcia\_Dissertation.pdf, 2022.

Molina García, V., Efremenko, D. S., and del Aguila, A.: Automatic differentiation for Jacobian computations in radiative transfer problems, Oral talk presented at 21st European Workshop on Automatic Differentiation, Friedrich Schiller University Jena, Jena, Germany, 2018.

In the simplest case from Tab.2, are you feeding the NN only with TOC for each class and let the hidden layers find a mapping between TOC and profile shape? It seems to me that in this case there could be profiles with the same TOC but different shape even within the same class. I just wonder how the NN is able to distribute the TOC variations vertically without having unique solutions.

 $\rightarrow$  It seems that the NN is able to deal with it, but the quality of the model using only TOC as input is worse compared to the models using additional input information.

How do you get to the number 420 in Table 2? I understand the 242 possible combinations in Table 3, as you have 2 hidden layers and 11 possibilities for each, times 2 options for the inputs, but I could not get to 420 combinations in Tab.1.

→ The number 420 in Table 1 results from 4 options for the number of hidden neurons (10,20,30,50) and the variation of the number of hidden layers (1,2,3). In case of 1 hidden layer this leads to  $4^1$  possibilities, in case of 2 hidden layers, we have  $4^2$  possibilities (e.g., [10,10],[10,20],...), and in case of 3 hidden layers, there are  $4^3$  possibilities (e.g., [10,10,10], but I tested also for example [30,50,30]). In total, there are 84 (=4+16+64) combinations times 5 options for the input (1st row in Table 1).

A side note: is the spiky shape of the profiles in Fig. 5 a feature of the RAL retrievals? Most of them tend to have three local maxima.

 $\rightarrow$  The jagged appearance in the figures is simply due to changing layer thicknesses.

**Technical corrections**

Line 13: I would add "presented in this manuscript" after "the homogenization".

 $\rightarrow$  Added as suggested.

Line24: "banned" → "prevented"

 $\rightarrow$  Changed.

Line 30-31: "the middle latitudes of the Northern Hemisphere"  $\rightarrow$  "at northern mid-latitudes"

 $\rightarrow$  Changed.

Line 59: Add a , after "data sets".

 $\rightarrow$  Comma added.

Lines ~60: You could mention the advantage/disadvantage to use limb or nadir data to retrieve profiles in terms of vertical and spatial resolution.

 $\rightarrow$  We added the following sentence after line 65:

"In contrast to limb satellite sensors, the nadir sensors benefit from a higher horizontal resolution and sensitivity also to the troposphere. On the other hand, their vertical resolution is limited."

Line 79: "allows us to generate"  $\rightarrow$  "enables the generation of"

 $\rightarrow$  Changed.

Line 80: "in particular important"  $\rightarrow$  "particularly important"

 $\rightarrow$  Changed.

Line 82: What is it meant with "investigation of changes in the profile"? Stratospheric ozone trends?

 $\rightarrow$  Yes. We have reworded this sentence:

"In addition to investigations of vertically-resolved trends,..."

Line 85: "enables us to assess"  $\rightarrow$  "facilitates the assessment of"

 $\rightarrow$  Changed.

Line 97: Add , after "ozone profiles".

 $\rightarrow$  Comma added.

Line 98: The UVN acronym was already introduced in the previous page.

 $\rightarrow$  Removed.

Line 160: Add , after "level-2 products".

 $\rightarrow$  Comma added.

Line 202: Also at northern mid-latitudes SCIAMACHY has a positive bias.

 $\rightarrow$  We have added the NH mid-latitudes in this statement.

Lines 205-207: I would move this last three lines to the beginning of the paragraph (line 194), as these are general considerations about the seasonal cycle.

 $\rightarrow$  Agreed.

Line 216: "drift" is repeated two times.

 $\rightarrow$  Corrected.

Lines 220-222: Do you plot in Fig. 2 the fit, for example, to (GOME-OMI) anomalies (as you state in the text) or to OMI-GOME?

 $\rightarrow$  We show the fit, e.g., OMI-GOME. The statement in the text and the figure caption was wrong and has been corrected.

Line 227-229: I find hard to read the sentence starting with "From these deviations...". I suggest to reformulate such as: "From the time series of the offsets in each available spatial bin, at first, we calculate averages for each calendar month ("climatologies") and then we average them over five broad latitude bands...".

 $\rightarrow$  We reformulated the sentence as suggested.

Line 242: "aligning" or "harmonizing"?

 $\rightarrow$  Changed to "harmonizing".

Line 262: "in particular as to the..."  $\rightarrow$  "in particular in terms of the..."

 $\rightarrow$  Changed.

Line 328: Remove , after "requires".

 $\rightarrow\,$  Comma removed.

Line 360: "of the parameters total ozone..."  $\rightarrow$  I would add "of the parameters, i.e. total ozone..."  $\rightarrow$  Agreed.

Line 371: Add , after "in advance"

 $\rightarrow$  Comma added.

Line 402: "only for example poleward of 50° N for 120°-180°"  $\rightarrow$  "mostly at latitudes poleward of 50° N and at 120°-180° E."

 $\rightarrow$  Changed.

Line 436: "measurements from"  $\rightarrow$  "measurements over"; "data from"  $\rightarrow$  "data over".

 $\rightarrow$  Changed.

I would remove Line 446 as it repeats what said in the previous lines.

 $\rightarrow$  Changed.

---

## Author Comment (AC2)

Reply to Anonymous Referee #2

We thank anonymous referee #2 for her/his helpful comments and corrections. Please find below the reviewer's comments (in black), our responses (in blue), and changes or additions to the text (in red).

Anonymous Referee #2, 27 Mar 2025

"The novel GOME-type Ozone Profile Essential Climate Variable (GOP-ECV) data record covering the past 26 years", by Coldewey-Egbers et al., presents a new data record of homogenized vertically resolved ozone data from nadir-viewing satellites that have been operating since the mid 1990's. The authors describe their homogenization methodology, which includes both inter-satellite harmonization and then homogenization to the existing GTO-ECV record, which is based on total ozone column measurements from a similar subset of satellites. Given the significant uncertainties and controversies around recent ozone trends in both the troposphere and stratosphere, this new homogenized record will be a helpful contribution to the community. A careful description and documentation of how these types of data records are constructed is both necessary and a good fit for the AMT journal. Overall, this article does a good job of describing the chosen methodology for constructing the GOP-ECV record. However, some of the methodological choices seem to be lacking a clear justification, and my questions/concerns around these choices form the basis for my major questions, which I outline below. Other than these issues, there are a number of generally minor grammatical/typo/clarity issues that I identify in the minor comments section following these more major concerns:

I understand the general motivation for wanting to create an ozone profile dataset whose vertically integrated column (i.e., TOC) matches that from an independent TCO dataset (which is believed to be stable and accurate for long term trends). But the number of adjustments being made to the profile dataset is concerning to me and confounds the interpretation of likely analyses with this data set. For example, each individual instrument is detrended and bias corrected relative to OMI as a function of longitude, latitude, and level, and then the entire merged GOP-ECV record is bias corrected to GTO-ECV. It is challenging for me to understand how this process distorts the original measurements, and whether or not the resulting GOP-ECV can really be trusted for trend studies or considered as an independent trend estimate given the way it is tied to GTO-ECV. One could imagine an alternative merging where the source trend records are preserved (i.e., only a bias offset is applied to homogenize the profile records). Then it would make sense to me that GOP-ECV could be considered an independent source for trends. With the described methodology, I think the interannual variability of the source records is preserved, which is scientifically useful, but it is not obvious how users of this data should interpret trends. Some discussion of this issue is critical to include, as it is highly likely that users will want to use this data set for ozone trend studies.

→ We understand the reviewer's concerns regarding the double adjustment applied to the profile data. However, we think that a bias correction and de-trending (instead of bias correction only) with respect to OMI data before the merging is preferred. We are aware of other merged profile records (e.g., SWOOSH, GOZCARDS, SBUV-NASA/-NOAA) which do not apply a drift correction before merging. In our case, an advantage of the series of nadir sounders is that all non-reference sensors have very long overlap periods with the reference OMI (at least 7 years), and all of them provide a very good spatiotemporal sampling (except for GOME after June 2003). We think that this enables the estimation of robust correction factors. Keppens et al. (2018) found in their validation study that L2 OMI and GOME profile drifts are overall insignificant. Altough this is not the case for the corresponding L3 data, even larger height-dependent drifts were found for SCIAMACHY and GOME-2A (GOME-2B was not part of the drift analysis). The authors indicate that appropriate corrections can make the data sets fit for climate applications. We further expect, that the second adjustment (the tying to GTO-ECV) then accounts for the remaining drift (mainly inherent to OMI). We included a short discussion in Sec. 3 of the manuscript:

"The decision to apply a time-dependent bias correction instead of a simple bias correction was mainly motivated by the findings of Keppens et al. (2018). Their validation of profile data with ground-based measurements yielded that strong height-dependent drifts were found in particular for the SCIAMACHY and the GOME-2A sensor. Smaller values were found for GOME and OMI, whereas GOME-2B was not yet part of this drift analysis. Since the overlap periods of the non-reference sensors with OMI are sufficiently long (at least 7 years) and the spatiotemporal coverage is very good for all instruments, too, an estimation of robust correction factors is probably feasible."

Regarding the reviewer's concerns about the independence of GOP-ECV: The motivation is to make GTO-ECV and GOP-ECV consistent in terms of the total column and we are aiming at retrieving the same (total column) trend from both records.

An obvious, and much simpler, method for scaling the merged profiles would be to apply a multiplicative scaling uniformly to the profiles so that their TOC matches that of GTO-ECV. Some better justification of the complicated method used here (i.e., clustering, classification, neural network to get Jacobian) is needed. By adopting the method used in this paper, the authors are implicitly acknowledging that there are altitude-dependent adjustments that should be made to the profile data, but the justification for why one would expect this to be necessary is not clearly stated. In addition to providing a better justification for the complicated altitude-dependent scaling algorithm applied here, it would seem simple enough to provide some information on how much different the altitude-dependent scaling is from a simple (altitude independent) scaling of the L3 profiles to match GTO-ECV.

→ The figures below show the mean difference between the scaled and the non-scaled profiles for both an altitude-dependent scaling and a uniform scaling. The top row shows the absolute difference and the bottom row indicates the difference in percent. We show the difference for 4 months in 2014 (January, April, July and October; from left to right). In each panel the mean difference is shown for the tropics (30°N-30°S, orange and red curves) and the middle latitudes (30°-60°, blue and purple curves, in the Northern Hemisphere for January and April (1st and 2nd column) and in the Southern Hemisphere for July and October (3rd and 4th column). The main findings are:

The difference in absolute values (DU) between both scaling methods is small for the tropics for all months (difference between the red and the orange curves in the top panels).

The difference in absolute values (DU) between both scaling methods is larger for the middle latitudes in all months (difference between the purple and the blue curves in the top panels). The difference is largest (up to 1-2 DU) in the 2nd layer (450-170hPa).

The difference between the altitude-dependent scaling (orange and blue) and the uniform scaling (red and purple) is also visible quite well for the percentage difference (bottom panels). In the lowermost layers, we found the largest discrepancies of up to 5% between the altitude-dependent and the uniform scaling, in particular for the middle latitudes (blue-purple curve). For January and April, the relative difference for the tropics (orange-red curve) is of the same order as the difference in the middle latitudes (blue-purple), whereas it is smaller for July and October.

We think that the results of this investigation justify the use of the altitude-dependent scaling method we propose.

On top of that, the Jacobians derived from the Neural Network approach depict the altitude-dependent correlation of total ozone and the profile shape, and they underpin that the relationship is not uniform. Moreover, work related to the generation of ozone profile climatologies that are binned according to

total ozone (e.g., Lamsal et al., 2004; Labow et al., 2015) indicates a non-uniform relationship between profile shapes for two neighboring total ozone bins.

Even if the proposed altitude dependent scaling method is justified, I'm concerned that the specific methodological choices made are not optimal, and some further explanation or exploration of sensitivity to methodological choices is needed. The described method seems to be a mixture of ML methods applied to both level 2 (i.e., individual ozone profile retrievals) and level 3 (monthly mean) data. For example, the clustering algorithm is applied to level 2 data, but then the classification is applied to L3 monthly mean merged profiles (e.g., Fig. 7) and used (in Sect 4.3) in the NN (at least, as far as I can tell this is what is happening. The description in Sect 4.3 about exactly what data is used in the NN is confusing and needs to be more specific about, e.g., which TOC is being used as input). If I'm understanding the description correctly, there is an assumption here that the L3 Jacobians should behave in the same way as the L2 Jacobians for a given class, but in reality the L3 values are in some (many?) cases a linear combination of multiple different classes of ozone profiles, so it's not obvious to me that the Jacobian methods developed with L2 data can be simply transferred to L3 data (or vice versa). Conceptually, it makes more sense to me to do the clustering/classification directly on the L3 profile data, and then train the NN with the L3 TOC on these classes (presumably there are fewer classes in L3 space?). Or, alternatively, do everything in L2 space (including the altitude-dependent scaling) before creating the L3 monthly means and merging. There may be justifications for mixing L2 and L3, but the mixing should be more clearly described and justified.

 $\rightarrow$  The entire approach described here, is done on the L3 data. There is no mixture between L2 and L3. The clustering, the classification and the calculation of the Jacobians use L3 as input. We have made some additions to the text (Sec. 2.2) in order to make this more clear and to avoid confusion.

Minor issues:

Line 2: Why does the dataset end in October 2021 when GOME-2 continues past this time?

 $\rightarrow$  There was a change in the GOME-2B processor after 2021. UV degradation to the instrument also increased in the later part of this mission. Moreover, OMI data is available on the CDS until October 2021 and we decided to limit the first version of GOP-ECV to this period.

Line 14-16: I don't necessarily doubt that this sentence is true (that GOP-ECV agrees with other long-term data records), but it has not really been demonstrated in this paper and is therefore inappropriate to include this assertion in the paper.

 $\rightarrow$  We rephrased the sentence.

"We found that the climatological ozone distributions derived for selected layers from the final GOP-ECV data record indicate expected spatiotemporal patterns."

Line 24: Instead of "banned" I think you mean "halted"?

 $\rightarrow$  "banned" has been replaced with "prevented". (See comment by reviewer #1)

Line 70: "An homogeneous" should be "A homogeneous"

 $\rightarrow$  Corrected.

Line 73 (and elsewhere): I don't think it is appropriate or necessary to abbreviate "with respect to".

 $\rightarrow$  Corrected (all occurrences in the manuscript).

Line 97: Bad grammar in this sentence. Could fix by changing "...climate data record of ozone profiles measurements..." to "...climate data record, ozone profile measurements..."

 $\rightarrow$  Changed as suggested.

Line 98: "are combined viz." is an awkward way to introduce a list. Suggest using a colon instead.

 $\rightarrow\,$  Replaced with a colon as suggested.

Line 111: I've never heard of a daytime. I think you mean local time.

 $\rightarrow$  Replaced with "times of day".

Line 129: It seems suboptimal to use v2 for OMI but v3 for the other four sensors, especially given that is the reference. What are the implications of this?

 $\rightarrow\,$  For all sensors we used the most recent version that was available.

Line 131-132: I'm confused by the reference here to you using L3 data from CDS. I thought you were constructing an L3 data set from the L2 profiles? This is alluded to on line 133.

 $\rightarrow$  We used the L3 data from the CDS as input. We rephrased this part in order to avoid confusion.

Line 132-133: What do these versions mean?

 $\rightarrow$  These version numbers refer to the version of the Level-3 data from the CDS.

Line 134-135: What do you mean "Monthly mean profile information is provided on a 1° x 1° grid"? I thought you were gridding the data. Maybe you are just describing what Keppens et al did, but this sentence is confusing and the information on their grid size doesn't seem relevant.

 $\rightarrow$  See reply to comment ,Line 131-132'. We have rephrased this part.

Line 159: The word "basically" seems ambiguous and unnecessary here.

 $\rightarrow$  Deleted.

Line 185: Shouldn't the left side of this equation be "sigma^2", not just "sigma". Or alternatively should there not be a square root on the righthand side?

 $\rightarrow$  The reviewer is correct. The left side should be " $\sigma^{2^{"}}$ . Corrected.

Figure 2: OMI seems to show a strong negative trend over the 2005-2010 time period which isn't present in GOME or SCIAMACHY. Is this real? What are the implications for this of using OMI as the reference?

 $\rightarrow$  Unfortunately, we cannot easily verify if the negative trend for OMI seen in this figure is real, because there are no ozone sonde stations available for that particular latitude/longitude grid cell. The analysis performed by Pope et al., 2023 (their Fig. 5) for the same layer (lower troposphere) but the entire tropical belt (30°S-30°N) yielded that the OMI data, which are based on the same retrieval algorithm as in our study, do not indicate a significant trend compared to sondes. On the other hand, SCIAMACHY data indicate significant positive deviations w.r.t. sondes for 2006-2009, and GOME-1 indicates significant negative deviations over 2002-2003.

**Line 215: "first order polynomial". Do you mean a linear fit?**

 $\rightarrow$  Yes. We have replaced it with "linear fit".

Line 217: Exemplarily is an odd word to use here.

 $\rightarrow$  "Exemplarily" has been deleted.

Line 258: "due the" should be "due to the"

 $\rightarrow$  Corrected.

Line 276: It's awkward to start the numbered list with "and"

 $\rightarrow$  Deleted.

Line 351: "ozon" should be "ozone"

 $\rightarrow$  Corrected.

Line 358: The NN is trained on the samples shown in Fig. 5, which are L2 profiles if I understand correctly. But then later the NN is applied to L3 data. How applicable is it to do this? See my major question above.

 $\rightarrow$  The training is also based on L3 data. See reply to major question no. 3.

Line 361: Which total ozone is being used as input here? Is it the integration of the 19 partial columns in each profile under consideration, or is it the TOC from GTO-ECV? Ultimately it is the GTO-ECV that is used as input for applying the altitude-dependent scaling (i.e., in eq. 7), so one would think that is what should be used for training. But I'm not sure that is even possible to do that for training. Some clarification is needed here.

→ The goal of using NNs is to find a (non-linear) mapping between total ozone and the ozone profiles, therefore we must use as input for the training the integrated column from the profiles. Note that using GTO-ECV columns in this step will induce an unwanted inconsistency. Once this mapping is obtained (after the NN training) we use GTO-ECV total columns for computing the NN-Jacobians needed for calculating the altitude-dependent scaling.

Line 370: "providing" should be "provides"

 $\rightarrow$  Corrected.

Line 398: Exemplarily is an odd word choice and could be removed.

 $\rightarrow$  Removed.

Figure 12: Why is there a longitudinal structure around Antarctica in the troposphere that seems to (somewhat) mimic the stratosphere? Is this present in the input data or an artifact of the altitude dependent scaling? Also, please change this figure to use a sequential color scale rather than a rainbow one (for reasons such as this, e.g., https://theconversation.com/how-rainbow-colour-maps-can-distort-data-and-be-misleading-167159).

 $\rightarrow$  The longitudinal structure in the troposphere is also present in the input data. The plots show the climatology for October for the lowermost layer (surface-450hPa). Both show the same longitudinal pattern. Left: GOP-ECV 5°x5° and right: original input OMI 1°x1° data from the CDS.

We now use a sequential colormap for Fig. 12.

(d) Partial ozone column (surface-450hPa) October (d) Partial ozone column (surface-450hPa) October

---

## Author Response (AR2)

Dear Troy Thornberry,

Thank you very much for the positive decision. We have considered all comments you raised (see below).

Kind regards, Melanie Coldewey-Egbers

Comments (line number refer to the revised manuscript with tracked changes):

L16 "indicate" might be better "exhibit" or possibly "reproduce" or "reflect" (also L503)

→ Replaced with "exhibit".

L127 suggest omit "that is" and "and"--"(RAL) scheme, a sequential three-step process described in detail"

→ Done.

L128 could include, perhaps parenthetically, the top level pressure since the retrieval is defined on fixed pressure levels

→ Top level pressure (0.01 hPa) has been included.

L133 comma--"For OMI, version 2 of the retrieval scheme is used" → Done.

L160 comma--"project, it is" and omit "it"--"basis and is freely" → Done.

L203 omit "the"--"strongly depends on latitude and altitude" → Done.

L220 comma--"(see Sec. 2.3), we use OMI"  $\rightarrow$  Done.

L228 should "polynomial" also be "linear" here in "the straight lines for...denote the results of the polynomial fits"?

→ Replaced with "linear".

L238 omit "probably"--you are arguing that it is indeed feasible since that is the approach you have taken, correct?

→ Deleted "probably".

L243 perhaps "spatial bin, we first calculate...and then average" → Changed as suggested.

L253 "+3 DU (May and June in the NH"--from the plot it looks more like June and July (?), but the trace colors are close...

→ Changed; June and July is correct.

L290 comma--"Sec. 3, thereby aiming" → Done.

L298 you ended up with a "3." before the ", and"

→ Seems to be an artifact of the "latexdiff" procedure to generate the version including track changes.

L384 "i.e." should either be omitted if the list is complete or "e.g." if the parameters listed only some

→ "i.e." has been deleted; the list is complete.

L416 should "the information to which class the profile" be just "the class to which the profile"?

→ Changed.

L459 perhaps "solely consists of GOME measurements (1995 - 2002) or of GOME and SCIAMACHY (2003 - 2004)"  $\,$

→ Changed as suggested.